# CellxPert: An Efficient Reasoning Language Model for Single-Cell & Spatial Multi-Omics

## Abstract

In this work, we introduce CellxPert, a scalable multimodal foundation model that unifies single-cell and spatial multi-omics within a common representation space. CellxPert jointly encodes transcriptomic (scRNA-seq), chromatin-accessibility (ATAC-seq), and surface-proteomic (CITE-seq) measurements, while directly incorporating MERFISH and imaging mass-cytometry data as 2D or 3D spatial–visual layers. CellxPert facilitates four key downstream tasks out of the box: (i) cell-type annotation across a broad ontology of 154 largely overlapping identities—the largest label space addressed to date and a stringent test of fine-grained discrimination, (ii) efficient fine-tuning using Low Rank Adaptation (LoRA), (iii) genome-wide transcriptomic response prediction to in silico perturbations (ISP), and (iv) seamless multi-omic integration across various assays and platforms. Unlike current single-cell foundation models, which approximate gene perturbations by deleting or reordering tokenized gene expression ranks, CellxPert employs a Metropolis–Hastings sampler whose proposal kernel uses the model's masked conditional distributions to transition to new transcriptomic states conditioned on the perturbed genes. This Markov-chain procedure mitigates out-of-distribution artifacts introduced by abrupt token manipulation and produces trajectories that are biologically interpretable. Evaluations on PBMC68K, Replogle Perturb-seq, Systema and BMMC benchmarks show CellxPert outperforming classical and state-of-the-art baselines in cell-type annotation, perturbation-aware reasoning, and multi-omic integration by a significant margin.

## 1 Introduction

Cellular systems generate heterogeneous, high-dimensional data spanning molecular, cellular, and tissue scales. To process this multimodal evidence, AI systems must jointly encode sequence information, quantitative expression profiles, and spatial context within a single latent space (Bunne et al., 2024; Heumos et al., 2023; Ashuach et al., 2023). We present CellxPert, a multimodal foundation model that constructs hierarchical representations of cell state by composing three abstraction layers: *molecular*, *cellular*, and *multicellular* that map complementary observables into a shared latent space. Specifically, (i) the *molecular* layer applies a sequence-aware transformer to DNA/RNA/protein tokens; (ii) the *cellular* layer forms a permutation-invariant set representation by additively fusing feature identity embeddings with expression magnitude encodings, and (iii) the *multicellular* layer builds a spatial neighborhood graph with relative positional encodings to capture tissue context. A provides a holistic view of the molecular, cellular, and multicellular abstractions.

Beyond observational inference, a central use case of single-cell foundation models is *reasoning* about *in silico perturbations* (ISP), which predict genome-wide responses to gene knockdown or overexpression to understand transcriptomic regulation and to validate therapeutic targets (Adamson et al., 2016; Dixit et al., 2016). However, most single-cell foundation models are trained solely on observational data, without exposure to perturbations during pretraining. To simulate gene knockdown or knockup, existing approaches typically delete or reorder tokenized gene expression ranks (Theodoris et al., 2023). Such simplistic manipulations neglect biological regulatory mechanisms and push the model into out-of-distribution regimes, resulting in unreliable embeddings for genome-wide perturbation analyses. An alternative is conditional masked imputation (Cui et al., 2024). Instead of directly rearranging the order of the ranks, this method fixes a subset of gene tokens and introduces tokens that encode experimental conditions, e.g., perturbation labels. The remaining positions are masked

and imputed in one forward pass by a masked-gene prediction head. While this keeps inference near the pretraining distribution and avoids explicit rank manipulations, it is susceptible to mode collapse regressing toward global expression baselines, diminishing biologically meaningful variability, and accuracy degrades under heavy masking (Kedzierska et al., 2025; Wu et al., 2024).

In practice, ISP pipelines modify tokenized representations for a target gene set and then measure the effect on internal embeddings, e.g., `[CLS]`, mean-pooled cell vectors, or per-gene embeddings. Typical operations include *deletion* (remove tokens to mimic underexpression), *overexpression* (move tokens forward to simulate higher rank/priority), and *activation/inhibition* (shift tokens across rank quantiles). Following perturbation, a forward pass on the altered token sequence generates updated embeddings. These are compared to the original embeddings via cosine similarity, with greater dissimilarity indicating a more pronounced shift in cell state. Despite their practicality, these token-based methods suffer from critical limitations:

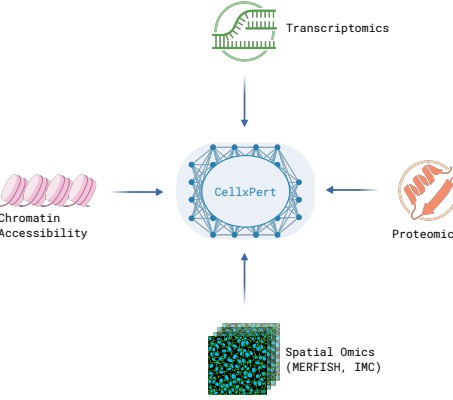

**Figure 1:** CELLXPERT is a **generalist agent** sharing a single backbone across modalities and tasks.

---

**Limitations of Existing Token-based Gene Perturbation Modeling**

**Oversimplification of Biological Mechanisms.** Simply deleting a gene token to mimic underexpression or repositioning it earlier in the ranking to simulate overexpression represents a crude approximation. Actual biological processes involve complex interactions and regulatory networks, which are not realistically captured merely by deleting or reordering tokens.

**Out-of-Distribution Embeddings.** Extreme perturbations, e.g., complete deletion or strong overexpression, may push the model into data regimes that were not encountered during training. The resulting altered token sequences yield unreliable embeddings, as the model must extrapolate to unseen configurations. This results in embeddings that are not trustworthy because the model is essentially guessing based on situations it never saw during training. Predictions in these scenarios may lead to misleading biological conclusions.

**Baseline Sensitivity.** These methods show sensitivity to the perturbation type, where genes identified as highly important to the model's predictions vary depending on whether the perturbation involves deletion, upregulation, or downregulation.

---

To overcome these challenges, we propose a Markov chain Monte Carlo (MCMC) wrapper for ISP within CELLXPERT. The method requires no supervision beyond class anchors computed on the training split. For each class, control or perturbed, we estimate a Fréchet medoid in embedding space. Here, perturbed denotes cells in which a designated gene is targeted by CRISPR. At inference time, users specify target genes and desired expression levels. We clamp these targets and evolve a Markov chain using masked language model (MLM) proposals. Each step randomly masks 15% of non-target genes, which matches our pretraining mask rate, then prompts the encoder to impute their values and applies a Metropolis–Hastings acceptance rule that favors transcriptomes whose cumulative per-gene distributions move toward the medoid of the perturbed class. This on-manifold, iterative procedure preserves biological variability, avoids distributional drift from hard token edits, and produces stable and interpretable trajectories of cell-state change.

## 2 RELATED WORK

Recent single-cell foundation models differ in architectural design, vocabulary scale, training objectives, and supported tasks as demonstrated in Tables 8a and 8b. SCBERT (Yang et al., 2022) employs an encoder-only low-rank Transformer over ∼16k gene tokens per cell, masking non-zero entries to learn gene co-expression structure. SCGPT (Cui et al., 2024) uses a decoder-only GPT-style Transformer trained on ∼33M cells with generative pretraining. GENEFORMER (Theodoris et al., 2023) adopts a bidirectional BERT-style encoder trained on ∼30M cells (extended to 95M), learning contextual gene representations and regulatory hierarchies. XTRIMOGENE (Gong et al.,

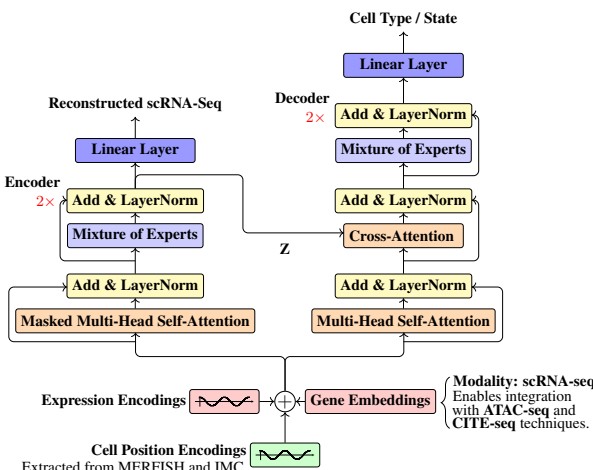

**Figure 2:** The overall architecture of CELLXPERT follows the encoder-decoder (seq2seq) paradigm.

2023) applies an asymmetric encoder–decoder architecture that skips sparse input tokens, scaling to 100M parameters and training on ∼50B gene tokens. CELLPLM (Wen et al., 2023) introduces a hierarchical model comprising a gene embedder, Transformer encoder, Gaussian mixture latent layer, and decoder, using cells as tokens and tissues as sentences. While most models use vocabularies of ∼20k–25k genes, CELLXPERT defines a 100k-token vocabulary by discretizing multi-omic signals and supports input sequences of up to 16384 tokens. These models vary in task specialization: SCGPT supports cell type classification, multi-omic integration, perturbation response modeling, and gene network inference. GENEFORMER is designed for gene function prediction, inferring gene–gene regulatory interactions, and modeling perturbation responses. XTRIMOGENE targets perturbation, drug synergy, and annotation tasks. CELLPLM incorporates spatial transcriptomics during pretraining, enabling joint modeling of scRNA-seq and spatial data for imputation and perturbation prediction. GEARS (Roohani et al., 2024), a specialized perturbation model, uses a graph-enhanced encoder–decoder architecture with gene–gene priors to predict transcriptional outcomes of single and combinatorial knockouts. A more comprehensive survey is provided in L.

## 3 SCALABLE MULTIMODAL ARCHITECTURE AND COMPONENT ABLATIONS

CELLXPERT is a two-stage encoder–decoder Transformer for multimodal single-cell inputs (gene expressions, DNA accessibility, protein abundance and cell coordinates) enhanced with mean sequence pooling, sparsely gated MoE feed-forward layers, and FlashAttention-v2. We pretrain the encoder with a masked-token objective on 4096-token sequences and then fine-tune the decoder for 154-way annotation using class-weighted cross-entropy. Long contexts near 4k tokens stress capacity and memory, so sparse MoE raises capacity at near-

**Table 1:** Ablations on 154-class annotation (Acc/F1 in %). *Ours* uses mean sequence pooling, MoE, FlashAttention-v2, 4096-token context, class-weighted cross-entropy.

| Variant | Ctx | Acc. | Macro-$F_1$ |
|---|---|---|---|
| Ours (mean, MoE, FA, wce) | 4096 | 69.2 | 63.8 |
| – Prepending `[CLS]` token | 4096 | 57.6 | 48.5 |
| – no MoE (dense FFN) | 4096 | 68.1 | 62.5 |
| – no FlashAttention | 4096 | 68.9 | 62.1 |
| – no class weighting | 4096 | 68.4 | 58.6 |
| Ctx = 1024 | 1024 | 65.7 | 60.4 |

constant per-token latency and FlashAttention reduces memory traffic, making long-context multimodal modeling feasible at fixed compute. Mean sequence pooling substantially outperforms a prepended `[CLS]` token. Table 1 shows that removing MoE or class weighting reduces both accuracy and macro-$F_1$, FlashAttention has negligible impact on quality (as expected from exact attention), and shortening context from 4096 to 1024 degrades performance. Additional details on the model architecture, routing, and scalability are provided in B and C.

We pretrained on 23.6M cells from the CELLxGENE Census (Program et al., 2025) (see E.1 for details). Inputs follow a compact preprocessing pipeline as presented in D: we perform quality control, per-cell 10k normalization, and log transform; select up to 4096 highly variable genes;

standardize by global per-gene mean and standard deviation; clip to two standard deviations; discretize expression into $B{=}50$ quantile bins; and tokenize each cell as aligned gene–bin pairs with fixed sinusoidal embeddings and optional spatial encodings. A unified gene vocabulary is maintained and extended during fine-tuning by appending unseen genes and growing the embedding table in place (see D.5). Hence, CELLXPERT supports incremental vocabulary growth across datasets, enabling *continuous learning* and seamless integration of unseen tokens and modalities during fine-tuning. Token-space augmentations further enforce permutation invariance (see D.6). We present each cell under independent random permutations of gene order; for Perturb-seq, where data are comparatively scarce and augmentation is essential, we apply minority oversampling, small bin jitter with clipping, and same-label token CutMix (Yun et al., 2019) while bin edges and statistics remain fixed. Pretraining minimizes cross-entropy only on masked positions, optimized with AdamW ($\beta_1 = 0.9$, $\beta_2 = 0.999$, weight decay $= 1 \times 10^{-2}$) under a cosine schedule with 1000 warm-up steps to a peak learning rate $1 \times 10^{-3}$ followed by decay to $1 \times 10^{-4}$. We use automatic mixed precision (AMP) with gradient scaling and distributed data parallelism (DDP). Fine-tuning follows a simple curriculum: 4 epochs with class-weighted cross-entropy, where the first 2 epochs use uniform weights and the final 2 enable inverse-frequency weights to progressively emphasize minority classes. Comprehensive pretraining details are provided in E.2.

## 4 SELF-SUPERVISED PRETRAINING EVALUATION

**Evaluation on Gene Expression Reconstruction** While many recent single-cell transformer models have adopted an MLM objective for pretraining, few have reported quantitative metrics for expression reconstruction, limiting direct comparisons in the field. To address this gap, we introduce the first rigorous, per-bin evaluation of MLM performance in single-cell transcriptomics. We discretize every gene's expression into 50 fixed bins and randomly mask 15% of these bin tokens across the input sequence. We then train the encoder-only Transformer to recover the original bin indices. We obtain a corpus-level accuracy of 96.3%, a macro-averaged $F_1$ of 96.0%. The confusion matrix in Figure 7 shows that most errors stem from assigning the highest expression bin to lower-expression categories, indicating that extreme outliers remain challenging.

**Large-Scale Cell- and Tissue-Level Discrimination** We evaluate CELLXPERT on classifying 154 cell types and multiple tissues, using a curated ontology to see if pretraining captures both fine- and coarse-level differences and follows the hierarchy. At the cell level, the model shows high performance on abundant, transcriptionally distinct lineages (e.g., hepatoblasts, retinal rod cells, oligodendrocytes), while errors concentrate among closely related or low-abundance subtypes (e.g., CD14/CD16 monocytes, NK maturation stages, CD4/CD8 $\alpha\beta$ T cells), consistent with shared or gradient markers (see confusion matrix in Figure 8). At the tissue level, accuracy is strong for transcriptionally stereotyped tissues (e.g., central nervous system, eye, embryo) and degrades primarily for anatomically or functionally overlapping systems (e.g., intestinal vs. mucosa; lung vs. respiratory), reflecting expected cross-tissue signature overlap (Figure 9). Full per-class and per-tissue metrics are provided in M (Table 9) and N (Table 10), respectively.

**CellxPert Outperforms Classical Baselines** We evaluate CELLXPERT using the PBMC68K dataset of peripheral blood mononuclear cells (PBMCs) (Zheng et al., 2017). We focus on 4 immune cell types that are functionally related and transcriptionally similar: CD8+ cytotoxic T (n = 15860), CD8+/CD45RA+ naive cytotoxic T (n = 13036), CD19+ B (n = 4460), and CD34+ progenitor cells (n = 180). These populations show overlapping gene expression and immune phenotypes, and their distribution is highly imbalanced, which makes them difficult to classify from RNA data alone. Prior work (Boiarsky et al., 2024) showed that logistic regression exceeds the performance of foundation models such as SCBERT on this benchmark. However, we find

**Table 2:** Class-imbalanced benchmark using PBMC68K (in %).

| Model | Acc. | $F_1$ |
|---|---|---|
| CELLXPERT | **78.1** | **78.9** |
| XGBOOST | 73.4 | 74.2 |
| RANDOM FOREST | 73.0 | 73.4 |
| L1-LOGREG | 68.2 | 69.5 |
| L2-LOGREG | 67.3 | 67.9 |
| SCANVI | 68.2 | 68.3 |
| PCA+KNN | 68.7 | 67.5 |

that CELLXPERT achieves significantly stronger results, reaching 78.9% macro-$F_1$ compared to 74.2% for the best classical baseline, XGBOOST. These gains suggest that CELLXPERT can capture fine-grained structure in transcriptomic space that is not as well recovered by linear or tree-based models.

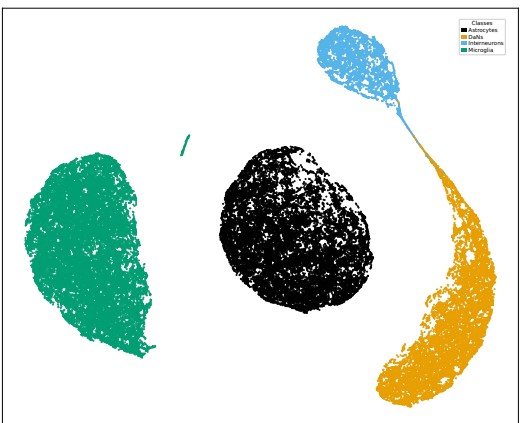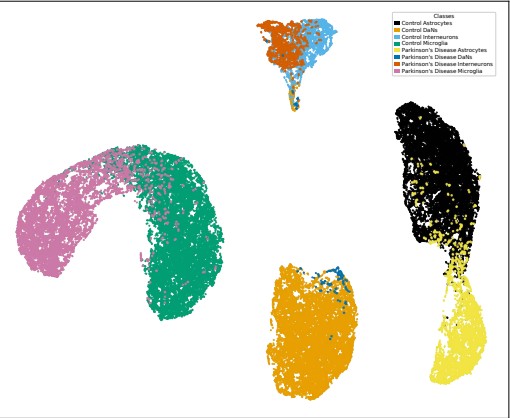

**Figure 3:** UMAP of fine-tuned CELLXPERT embeddings. **Left:** Colored by expert-annotated cell types; astrocytes, dopaminergic neurons (DaNs), interneurons, and microglia form distinct clusters. **Right:** Colored by disease status (PD vs. control).

**Table 3:** Macro-Averaged Performance for Cell Type Annotation & PD vs. Control Classification (metrics in %).

| Cell Type | Samples | Cell Type Annotation | | | | Control vs. Parkinson's | | | |
|---|---|---|---|---|---|---|---|---|---|
| | | Precision | Recall | $F_1$ | Accuracy | Precision | Recall | $F_1$ | Accuracy |
| All Cell Types | 13,113 | 99.7 | 99.6 | 99.6 | 99.6 | 82.7 | 74.9 | 74.9 | 74.9 |
| Astrocytes | 4,095 | 100.0 | 100.0 | 100.0 | 100.0 | 88.5 | 92.9 | 90.2 | 92.9 |
| DaNs | 2,752 | 99.5 | 99.6 | 99.6 | 99.6 | 73.3 | 56.9 | 60.6 | 56.9 |
| Interneurons | 1,121 | 99.1 | 98.8 | 98.9 | 98.8 | 75.5 | 58.5 | 56.5 | 58.5 |
| Microglia | 5,145 | 100.0 | 100.0 | 100.0 | 100.0 | 93.4 | 91.2 | 92.2 | 91.2 |

**Fine-tuning on Parkinson's Disease (PD)** We expect the value of single-cell foundation models to become even more apparent under distribution shift. Hence, we fine-tune CELLXPERT on the dataset from (Kamath et al., 2022) using a leave-one-control/one-PD-donor-out cross-validation protocol to mitigate donor-, tissue-, and batch-level confounders (Babcock et al., 2021; Tran et al., 2020). All transforms with leakage risk, including gene filtering, normalization, scaling, and highly variable gene selection, are fit per fold on training donors only. We use parameter efficient adapters by injecting low rank updates into attention and feed forward projections (implementation details are in G.1). We report macro averaged precision, recall, $F_1$, and accuracy on held out donors we aggregate across folds in Table 3. CELLXPERT maintains near ceiling cell type annotation where $F_1$ is 99.6%, while PD vs. control is more challenging where $F_1$ is 74.9%. Performance is higher for non-neuronal lineages such as astrocytes where $F_1$ is 90.2% and microglia where $F_1$ is 92.2%. Performance is lower for neuronal subtypes including dopaminergic neurons (DaNs) where $F_1$ is 60.6% and interneurons where $F_1$ is 56.5%. These trends align with the UMAP of cell embeddings in Figure 3 where major cell types form distinct clusters and disease status shows minimal overlap. These results remain strong under the weak supervision typical of single-cell disease datasets, where every cell from a patient inherits the donor-level diagnosis (Craig et al., 2025). The model also accurately highlights the glial populations most affected in PD, with robust signals from reactive astrocytes and microglia (Smajić et al., 2022).

## 5  METROPOLIS-HASTINGS SAMPLING FOR ISP RESPONSE PREDICTION

A common ISP baseline clamps the perturbed genes and then imputes the remaining positions in a single pass under a mean-field factorization. This factorization assumes conditional independence across masked genes and the formal definition is provided in H.1. In practice this procedure masks more than 99% of tokens at inference time, which is far outside the pretraining corruption rate. The result is distribution shift that collapses diversity and washes out gene to gene dependencies. Empirically, this yields centroid-like reconstructions and poor perturbation specificity, consistent

with prior work for SCGPT and also for non-transformer approaches such as CPA when evaluation emphasizes average shifts (Cui et al., 2024; Lotfollahi et al., 2023). Standard shift-sensitive metrics, such as Pearson correlation on differentially expressed genes or RMSE/MAE, can further inflate apparent performance because they reward global control to perturbation shifts while downweighting structure (Viñas Torné et al., 2025). Under such metrics simple linear baselines can match or surpass recent deep models on unseen perturbations (Ahlmann-Eltze et al., 2025; Csendes et al., 2025).

We therefore replace one-shot imputation with an MCMC sampler based on the Metropolis-Hastings algorithm (Hastings, 1970). At each iteration we randomly mask a small block of non-target genes that matches the pretraining corruption rate. We run the encoder to propose replacements conditioned on the unmasked context. We accept the joint update with a probability that increases as the candidate transcriptome moves toward anchors computed from perturbed cells. This keeps inference near the training manifold and preserves gene to gene dependencies. We monitor convergence by tracking the acceptance rate, as illustrated in Figure 11, and the stabilization of the mean per-gene expression shift between successive iterations. We model the BERT-like MLM in CELLXPERT as a fully connected Markov random field over genes (Devlin et al., 2019; Wang & Cho, 2019). The encoder logits define positive potentials and masked token training maximizes a tractable pseudo likelihood. The negative sum of site wise logits defines an energy over transcriptomes and induces a Gibbs distribution. This view is compatible with the use of local conditional proposals in our MCMC inference. We provide formal definitions and theoretical details in H.2.

**Preliminaries.** We frame the prediction of transcriptional responses to genetic perturbations as a Bayesian inference problem, which we solve using Metropolis-Hastings applied to expression profiles from control cells. The MCMC chain draws samples from a proposal distribution and accepts proposals that shift toward regions of high joint probability (biologically plausible states). Specifically, at each iteration we (i) clamp user-specified target genes to their desired levels, (ii) randomly mask a small block of non-target genes to match the pretraining corruption rate, and (iii) prompt the CELLXPERT encoder to propose new expression values from the learned conditional distribution. This partial, iterative resampling keeps the chain close to the training manifold, preserves gene-gene dependencies and avoids the fill-all-at-once extreme masking while enforcing perturbation constraints, e.g., knockdown, knockout, overexpression.

Let $\mathsf{x} = \{0, 1, \ldots, B-1\}$ be the set of expression bins. Given a sequence $\mathbf{x} = (x_1, \ldots, x_L)$, where each $x_i \in \mathsf{x}$ denotes the discretized expression bin for gene $i$, with user-clamped targets $\mathcal{T} \subseteq \{1, \ldots, L\}$, at iteration $t$ we sample a block $M_t \subseteq \{1, \ldots, L\} \setminus \mathcal{T}$ with $|M_t| = k = \max(1, \lfloor pL \rfloor)$, where $p \in (0, 1)$ is the mask ratio. This block $M_t$ is sampled uniformly at random from all possible subsets of the appropriate size, ensuring that the proposal process remains unbiased and covers the sequence space effectively. Additionally, the masking block sampled does not intersect with the frozen target set $\mathcal{T}$, preserving the integrity of the fixed positions during the sampling process. $\mathbf{x}_{-M_t}$ denotes $\mathbf{x}$ with entries in $M_t$ masked.

**Target distribution to score new transcriptomic states.** We score each candidate expression profile by its gene-wise Wasserstein-1 distance to a small set of perturbed anchors constructed from the training split only, so test cells never contribute to the target distribution and there is no leakage across the train–test boundary. For each perturbation, we compute Fréchet medoids in the training set and retain the top $K = 5$ medoids as anchors, chosen heuristically as a balance between robustness and computational cost. We then precompute per-gene cumulative distributions around these anchors and evaluate the Wasserstein-1 cost in linear time in the number of genes and bins. We use Wasserstein-1 because it is an optimal transport distance that is sensitive to shifts in the full distribution of expression, not just changes in the mean. Full details are given in H.3.

**Proposal distribution and acceptance rule.** The encoder-side MLM head outputs logits $\phi_i(\mathbf{x}_{-M_t}) = [\phi_{i,v}(\mathbf{x}_{-M_t})]_{v \in \mathcal{X}} \in \mathbb{R}^B$, where $B$ represents the number of discretized bins for gene expressions and $v$ indexes a specific bin. Temperature $\tau > 0$ controls the balance between exploration and exploitation in the proposal distribution. When $\tau > 1$, the distribution flattens, which encourages more exploration by making less probable outcomes more likely. Conversely, when $\tau < 1$, the distribution sharpens, which promotes more exploitation by favoring the most probable outcomes. We use $\tau = 2$ in our implementation. Appendix H.4 reports an ablation over $\tau$ demonstrating that ISP performance is stable across a broad range of temperatures and improves gradually as $\tau$ increases

from very small values. We define the proposal distribution for each position as

$$q_i(v \mid \mathbf{x}_{-M_t}; \tau) = \frac{\exp(\phi_{i,v}(\mathbf{x}_{-M_t})/\tau)}{\sum_{b \in \mathcal{X}} \exp(\phi_{i,b}(\mathbf{x}_{-M_t})/\tau)}, \quad v \in \mathcal{X}.$$

We propose a new state $\mathbf{x}'$ by resampling only the positions within the masked block $M_t$: specifically, $x_i' \sim q_i(\cdot \mid \mathbf{x}_{-M_t}; \tau)$ for each $i \in M_t$, while keeping $x_j' = x_j$ for all $j \notin M_t$. Conditioning on the chosen block $M_t$, the block proposal densities are expressed as

$$q(\mathbf{x}' \mid \mathbf{x}, M_t; \tau) = \prod_{i \in M_t} q_i(x_i' \mid \mathbf{x}_{-M_t}; \tau) \quad \text{and} \quad q(\mathbf{x} \mid \mathbf{x}', M_t; \tau) = \prod_{i \in M_t} q_i(x_i \mid \mathbf{x}'_{-M_t}; \tau).$$

The forward pass queries the CELLXPERT encoder once on the masked input $\mathbf{x}_{-M_t}$ to obtain the set of proposal distributions $\{q_i(\cdot \mid \mathbf{x}_{-M_t}; \tau)\}_{i \in M_t}$ and to sample the proposed values $x_i'$ for $i \in M_t$. The reverse pass reuses the same block $M_t$ and queries the CELLXPERT encoder on the masked proposed input $\mathbf{x}'_{-M_t}$ to obtain $\{q_i(\cdot \mid \mathbf{x}'_{-M_t}; \tau)\}_{i \in M_t}$, which are then used to evaluate the probability of the original expressions $\{x_i\}_{i \in M_t}$. These forward and reverse passes together enable the computation of the Hastings ratio $q(\mathbf{x} \mid \mathbf{x}', M_t; \tau)/q(\mathbf{x}' \mid \mathbf{x}, M_t; \tau)$ in closed form as a product over the masked sites, which compares how likely the proposal would regenerate the original expressions and corrects for proposal asymmetry penalizing moves that are hard to reverse and rewarding those that are easy. The Metropolis-Hastings acceptance probability per batch element, conditioning on $M_t$, is

$$\log r(\mathbf{x} \to \mathbf{x}' \mid M_t) = (\log \pi(\mathbf{x}') - \log \pi(\mathbf{x})) + \sum_{i \in M_t} \left[ \log q_i(x_i \mid \mathbf{x}'_{-M_t}; \tau) - \log q_i(x_i' \mid \mathbf{x}_{-M_t}; \tau) \right]$$

For numerical stability, we compute the log-ratio $\log r(\mathbf{x} \to \mathbf{x}' \mid M_t)$, then set $\log \alpha(\mathbf{x} \to \mathbf{x}' \mid M_t) = \min\{0, \log r(\mathbf{x} \to \mathbf{x}' \mid M_t)\}$, and accept the proposal if and only if $\log u \le \log \alpha(\mathbf{x} \to \mathbf{x}' \mid M_t)$ for a uniform random variable $u \sim \mathcal{U}(0, 1)$. Each iteration of this process uses exactly two MLM forward passes (forward on $\mathbf{x}_{-M_t}$ and reverse on $\mathbf{x}'_{-M_t}$).

**Evaluation on Genome-Wide Perturb-seq Datasets.** We evaluate on the Replogle–Weissman Perturb-seq dataset (Replogle et al., 2022), which is available from the Gene Expression Omnibus under accession code GSE146194. The dataset provides $\sim 1.72 \times 10^5$ scRNA-seq profiles per cell line (K562 and RPE-1) with CRISPRi labels for 1092 and 1543 single-gene perturbations, respectively, plus $\sim 2500$ controls per line. We retain perturbations passing a two-stage filter: (i) on-target repression with $\log_2 \mathrm{FC} \le -1$ and Benjamini–Hochberg adjusted $p \le 0.05$ (two-sided vs. controls), and (ii) significant separability from controls by an energy-distance (E-distance) test. This yields $224/1092$ targets in K562 and $66/1543$ in RPE-1. For each target in the high confidence set we form three groups of cells. The groups are control cells, cells receiving CRISPR (perturbed), and ISP predictions generated from control cells.

**Silhouette with cosine distance.** We quantify separability between perturbed and control cells using the standard silhouette coefficient. Let $d(\mathbf{x}, \mathbf{y}) = 1 - \mathbf{x}^\top \mathbf{y}$ for unit-norm embeddings. For a cell $i$ with embedding $\mathbf{z}_i$ and label $y_i \in \{\text{control}, \text{perturbed}\}$, we define $a(i) = \frac{1}{|\mathcal{C}(y_i)| - 1} \sum_{j \in \mathcal{C}(y_i) \setminus \{i\}} d(\mathbf{z}_i, \mathbf{z}_j)$ and $b(i) = \frac{1}{|\mathcal{C}(\bar{y}_i)|} \sum_{j \in \mathcal{C}(\bar{y}_i)} d(\mathbf{z}_i, \mathbf{z}_j)$, where $\mathcal{C}(y)$ is the index set of cells in class $y$ and $\bar{y}_i$ is the opposite class. The per-cell silhouette is $\mathrm{sil}(i) = \frac{b(i) - a(i)}{\max\{a(i), b(i)\}} \in [-1, 1]$. The per-target silhouette is the mean of $\mathrm{sil}(i)$ over all cells from the control and perturbed groups for that target. The dataset level score is the mean over targets. Positive values show a cell is closer to its own class than the other, with larger values reflecting stronger separation.

**Cosine shift toward perturbation.** We quantify alignment of ISP predictions to the perturbed state using mean pairwise cosine similarity between $\ell_2$-normalized model embeddings. For groups $A, B \in \{\text{control}, \text{perturbed}, \text{ISP}\}$, we define $s_{\mathrm{pair}}(A, B) = \frac{1}{|A| |B|} \sum_{\mathbf{z} \in A} \sum_{\mathbf{z}' \in B} \mathbf{z}^\top \mathbf{z}'$, which averages cosine similarities over all pairs of cells across the two groups. For each target, we compute $\Delta = s_{\mathrm{pair}}(\text{ISP}, \text{perturbed}) - s_{\mathrm{pair}}(\text{control}, \text{perturbed})$. We report the fraction of targets with $\Delta > 0$, which measures how often ISP predictions are more similar to perturbed cells than the original controls. Empirically, blockwise Metropolis–Hastings sampling yields higher-fidelity ISP samples. As shown in Table 4, CELLXPERT achieves stronger alignment to perturbed ground truth than larger baselines while using fewer pretraining cells and a similar context length: with 23.6M cells

**Table 4:** Evaluation on the Replogle–Weissman Perturb-seq high-confidence subsets. Metrics: mean silhouette across perturbations and "+Shift" (count and fraction of targets whose cosine similarity shifts toward the perturbed condition).

| Model | K562 ($n$=224) | | | RPE-1 ($n$=66) | | |
|---|---|---|---|---|---|---|
| | Silh. (↑) | +Shift $n$ (↑) | +Shift % (↑) | Silh. (↑) | +Shift $n$ (↑) | +Shift % (↑) |
| CELLXPERT | 0.58 | **212** | **94.6** | **0.63** | **64** | **97.0** |
| GENEFORMER-2 | **0.61** | 181 | 80.8 | 0.60 | 58 | 89.0 |
| GENEFORMER | 0.34 | 65 | 29.0 | 0.45 | 31 | 47.0 |
| scGPT | 0.52 | 35 | 15.6 | 0.57 | 17 | 25.8 |

and a 4096-token context, it reaches 94.6% positive cosine shift in K562 vs. 80.8% for GENEFORMER-2 trained on 95M cells at comparable silhouette (0.58 vs. 0.61). On RPE-1, CELLXPERT attains 97.0% positive shift with a silhouette of 0.63, whereas token-editing and one-shot methods such as GENEFORMER and scGPT show weaker separability and substantially lower target-directed shift.

**SYSTEMA benchmark: moving beyond "too-good-to-be-true" Pearson scores.** Most benchmarks compute a differential expression profile for each perturbation $X$ as $\Delta_X^{\text{ctrl}} = \mu_X^{\text{pert}} - \mu^{\text{ctrl}}$, where $\mu_X^{\text{pert}}$ denotes the mean expression of cells perturbed at $X$ and $\mu^{\text{ctrl}}$ is the mean expression of control cells. Pearson-$\Delta$ then measures the correlation between the true and predicted $\Delta_X^{\text{ctrl}}$ across genes. Under this setup, a simple perturbed-mean baseline already captures most of the global shift from control to perturbed cells, and SYSTEMA (Viñas Torné et al., 2025) shows that this linear baseline can outperform or match deep models such as CPA, GEARS, and scGPT. In other words, models are largely rewarded for reproducing the average control-to-perturbed shift, not the perturbation specific component. Biologically, this global shift reflects that certain programs such as cell death, cell cycle, and blood-cell differentiation tend to be more active in perturbed cells, and many different perturbations push cells into similar stressed or dying states. This shared stressed phenotype creates a strong global difference between control and perturbed populations, which in turn can inflate apparent model performance when metrics are dominated by this global effect rather than by truly perturbation specific responses. SYSTEMA adopts a stricter reference that removes the global perturbation effect. Instead of centering on controls, it defines $\Delta_X^{\text{pert}} = \mu_X^{\text{pert}} - \mu^{\text{all pert}}$, where $\mu^{\text{all pert}}$ is the mean over all perturbed cells pooled across targets. Pearson-$\Delta$ is then recomputed using $\Delta_X^{\text{pert}}$, so a model must explain the perturbation-specific deviation from the global perturbed state rather than merely reproducing the overall control-to-perturbed shift. Under this setting, most methods experience a dramatic drop in correlation, many scores are near zero or even negative, and the perturbed-mean baseline is no longer competitive. scGPT is often among the strongest models on the SYSTEMA benchmark, but its gains are modest and several methods remain close to random performance. On this stricter evaluation, CELLXPERT achieves a substantially larger improvement in perturbation-specific prediction. On this stricter benchmark, CELLXPERT shows a highly substantial improvement in perturbation specific prediction. On Replogle K562, it achieves a mean Pearson-$\Delta_{\text{ctrl}}$ of 0.66 and a Systema-style Pearson-$\Delta_{\text{pert}}$ of 0.45. On Replogle RPE-1, it reaches 0.72 and 0.46, respectively. These results show that CELLXPERT not only recovers the global control-to-perturbed shift but also explains a larger fraction of the target-specific residual signal than previously reported baselines in the SYSTEMA benchmark, moving ISP evaluation closer to perturbation-aware expression modeling.

**Table 5:** SYSTEMA ISP benchmark in expression space on Replogle K562 and RPE-1. We report mean Pearson-$\Delta$ with control reference (standard) and with the SYSTEMA perturbed reference.

| Model | Pearson $\Delta$ (Standard) (↑) | | Pearson $\Delta$ (SYSTEMA) (↑) | |
|---|---|---|---|---|
| | K562 | RPE-1 | K562 | RPE-1 |
| CPA | 0.06 | 0.10 | 0.05 | 0.08 |
| GEARS | 0.22 | 0.48 | 0.00 | 0.19 |
| scGPT | 0.27 | 0.51 | 0.06 | 0.13 |
| Perturbed mean | 0.32 | 0.55 | 0.06 | 0.08 |
| CELLXPERT | **0.66** | **0.72** | **0.45** | **0.46** |
| *Gain over next best* | *+0.34* | *+0.17* | *+0.39* | *+0.27* |

**Table 6:** Inference strategy ablation on CELLXPERT for ISP response prediction.

| Inference method | K562 (*n*=224) | | RPE-1 (*n*=66) | |
|---|---|---|---|---|
| | +Shift *n* (↑) | +Shift % (↑) | +Shift *n* (↑) | +Shift % (↑) |
| Metropolis–Hastings (MCMC) | **212** | **94.6** | **64** | **97.0** |
| One-shot masked imputation | 171 | 76.3 | 47 | 71.2 |
| Token editing (rank/bin reorder) | 140 | 62.5 | 41 | 62.1 |

**Ablation on ISP Inference Strategies**   To isolate the effect of the inference strategy on ISP response prediction from the underlying model architecture and training procedure, we perform a detailed ablation comparing three decoding strategies applied to the same pretrained CELLXPERT model: (i) our iterative blockwise Metropolis–Hastings sampling, (ii) one-shot MLM imputation, and (iii) deterministic token editing that reorders discretized gene expression ranks/bins. As shown in Table 6, Metropolis–Hastings sampling provides substantially larger target-directed cosine shifts. On K562 it achieves 94.6% positive shifts, a gain of +18.3% over one-shot MLM imputation and +32.1% over token editing. On RPE-1 it reaches 97.0%, improving by +25.8% and +34.9%, respectively. These ablations show that our gains in ISP response prediction are driven primarily by the Bayesian inference via MCMC sampling, rather than by changes to the model architecture or training procedure.

## 6   SPATIAL AND MULTI-OMIC INTEGRATION

**Spatial transcriptomics and proteomics.**   We evaluated CELLXPERT on two distinct spatial datasets, which include 3D MERFISH from mouse brain tissue and 2D imaging mass cytometry (IMC) from breast tumors using UMAP of CELLXPERT embeddings, mesoscale patterns (neighborhood enrichment), and multi-scale spatial clustering (variograms via Moran's *I*). Dataset details are provided in I.1 and I.2. All spatial statistics use model predicted labels. The results in Figure 12 and I.3 show distinct clusters for major cell types, coherent tissue modules such as neurovascular and immune stromal interfaces, and variograms that decay smoothly with radius which indicates strong local clustering that weakens at larger scales. Figure 4 further highlights coherent tissue modules from neighborhood enrichment. In MERFISH we observe a neurovascular module, strong self enrichment of Ependymal, and staged Oligodendrocyte compartments. In IMC we observe compact self enriched epithelial tumor patches and a clear Macrophage and T Cell interface with Vimentin High Stromal

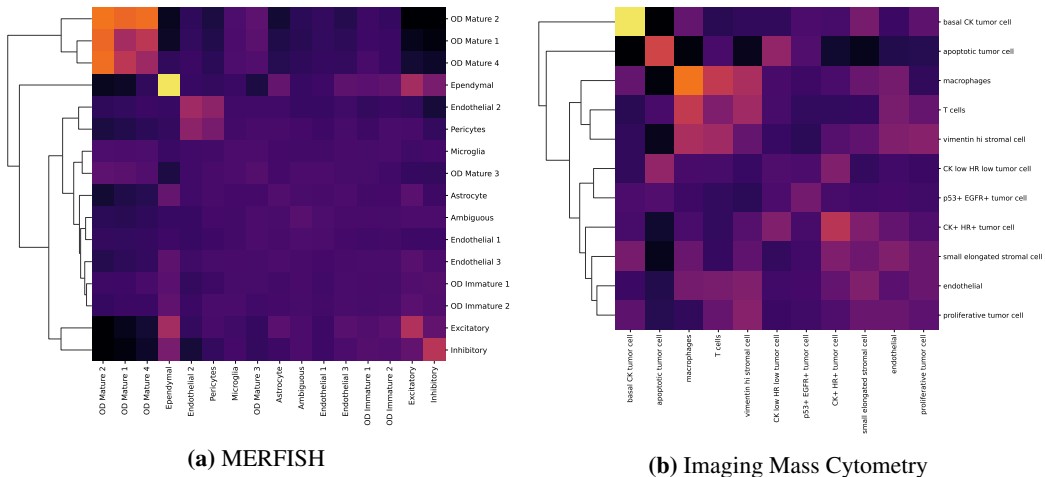

**(a)** MERFISH   **(b)** Imaging Mass Cytometry

**Figure 4:** Neighborhood enrichment from CELLXPERT predictions on MERFISH and IMC, computed from predicted labels to capture classifier-implied spatial organization. In MERFISH we observe strong self enrichment of *Ependymal*, co-enrichment of *Endothelial 2* with *Pericyte*, and *Oligodendrocyte* compartments that separate into immature and mature blocks with moderate cross links, which is consistent with a staged lineage. In IMC we observe compact self enriched patches formed by epithelial tumor cells, a clear *Macrophage* and *T Cell* interface with *Vimentin High Stromal Cell*, and depletion near *Apoptotic Tumor Cell*.

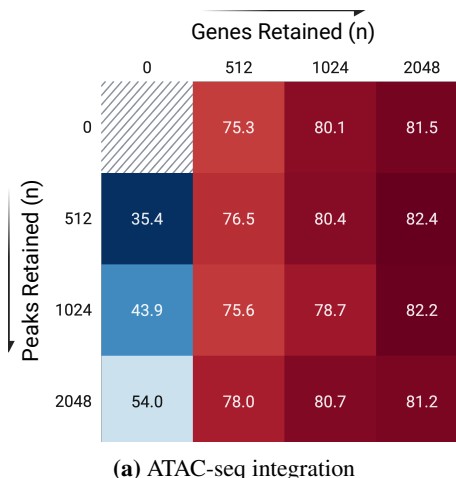

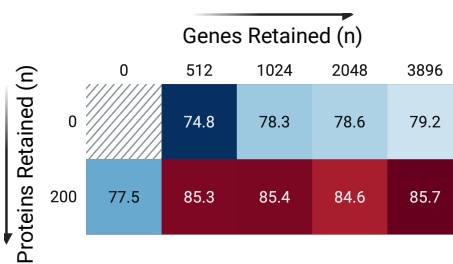

**(a)** ATAC-seq integration    **(b)** CITE-seq integration

**Figure 5:** Modality and token budget ablations. Test-1 accuracy (%) of CELLXPERT on BMMC as a function of token budget for (a) ATAC-seq and (b) CITE-seq. Columns denote the number of genes and rows denote the number of peaks or proteins. For each configuration, we retain the most statistically variable genes, peaks, and proteins within each modality.

Cell. Spatial metrics align with classification accuracy. Classes in compact niches achieve high $F_1$ and strong enrichment/autocorrelation signals. Examples include Ependymal at 91% and Endothelial 1 at 87% in MERFISH, and Basal CK Tumor Cell at 80% and Macrophage at 87% IMC. Errors concentrate in rare or transitional classes such as oligodendrocyte maturation stages and overlapping epithelial stromal programs. Class imbalance explains the gap between macro and weighted scores with MERFISH macro-$F_1$ at 61% vs. weighted $F_1$ at 77% and IMC macro-$F_1$ at 60% vs. weighted $F_1$ at 78%. A hierarchical evaluation that merges closely related subtypes could stabilize spatial metrics. Overall CELLXPERT provides robust and interpretable spatial representations across modalities.

**Joint modeling of gene expression, chromatin accessibility, and cell surface protein abundance** We evaluate on the OpenProblems NeurIPS 2021 BMMC benchmark under GEO GSE194122. We describe preprocessing, tokenization, and modality specific quality control for ATAC-seq and CITE-seq in J.1 and J.2, respectively. As shown in Table 7, under the same split and evaluation protocol CELLXPERT achieves 85.7% Test-1 accuracy and 86.3% weighted $F_1$, exceeding multimodal baselines by wide margins: +8.2% / +9.6% over SCMAMBA, +12.8 / +16.3 over SCGPT, and +19.4 / +21.7 over CELLPLM. To probe the sources of these

**Table 7:** NeurIPS 2021 BMMC benchmark (in %).

| Model | Acc. | $F_1$ |
|---|---|---|
| CELLXPERT | **85.7** | **86.3** |
| SCMAMBA | 77.5 | 76.7 |
| SCGPT | 72.9 | 70.0 |
| CELLPLM | 66.3 | 64.6 |

gains, we analyze modality and token–budget ablations. Figure 5 shows three consistent findings. First, RNA is the strongest single modality, ATAC alone is weaker but complementary, and ADTs alone are competitive because surface markers capture immune identity. Second, early fusion yields most of the performance gain. Adding 200 ADTs to 3896 genes raises Test-1 accuracy from 79.2% to 85.7%. It also yields large class level gains for receptor and activation defined states. For example *CD4+ T Activated* increases from 45.6% with RNA to 85.3% with the mixed model. Third, adding ATAC peaks to RNA helps at low gene budgets and then saturates. At 512 genes, 2048 peaks improve accuracy by 2.7%, while gains are negligible or negative at larger gene budgets. The gains from ADTs concentrate in fine grained T, NK, and myeloid phenotypes, and expanding the RNA vocabulary beyond roughly one to two thousand genes shows diminishing returns once a second modality is present. We provide detailed results on gains from multimodal fusion and token budget sweeps in J.3.

## 7 CONCLUSION

We present CELLXPERT, a multimodal foundation model for single-cell and spatial omics with a block Metropolis–Hastings ISP sampler that treats MLM as an implicit energy-based model to generate on-manifold transcriptomes. This preserves gene dependencies, avoids out-of-distribution shifts, and outperforms strong baselines on perturbation response prediction and multi-omic integration.

## REPRODUCIBILITY STATEMENT

All datasets used in our experiments are publicly available, with direct links and complete preprocessing scripts included in our codebase. Upon acceptance, we will release code, configuration files, checkpoints and a detailed README with setup instructions. This release will be hosted on GitHub under a non-commercial license.

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

## A    HIERARCHICAL ABSTRACTIONS INTO A SHARED LATENT SPACE

> **Model Abstraction**
>
> **Molecular layer.** RNA, DNA, and protein sequences are serialized into a flat sequence of tokens and processed with a stack of transformer layers. Although small molecule vocabularies (glycans, lipids, metabolites) are critical for complete biochemical coverage, their combinatorial diversity inflates the token set into the $10^5$–$10^6$ range, making even memory-efficient kernels such as FlashAttention (Dao et al., 2022) (with $\mathcal{O}(n^2 d)$ time complexity, where $n$ is the sequence length and $d$ is the per-head embedding dimension) computationally prohibitive on today's hardware.
>
> **Cellular layer.** Each omics token, whether it represents a gene, an ATAC-seq k-mer, or a CITE-seq epitope, is mapped twice: first to an identity embedding that captures what the feature is, and second to a magnitude encoding that captures how much of it is present after the raw measurement is discretized (e.g., binned expression levels, accessibility intensities, or antibody counts). Both identity embeddings and magnitude encodings share the same dimensionality, so we fuse them with a simple element-wise addition to produce the final token vector. This additive operation keeps the model's memory and compute footprint constant, unlike vector concatenation, while still allowing gradients to flow back to embeddings. It treats the two pieces of information as complementary channels of the same feature, similar to how transformers add positional encodings to word embeddings, and it gives the network freedom to learn whether identity or magnitude should dominate by adjusting the respective embedding weights during training.
>
> **Multicellular layer.** CELLXPERT brings in 3D transcriptomic spots from MERFISH and 2D proteomic cells from Imaging Mass Cytometry (IMC), then shift each axis so that all coordinates run from zero up to their original range. The resulting float tensors of each cell's $(x, y, [z])$ coordinates are run through a fixed sinusoidal encoder—matching the transformer's embedding dimension—and multiplied by a single learnable scalar before being added element-wise to every token's identity-plus-magnitude vector (Vaswani et al., 2017; Gehring et al., 2017). This approach incorporates spatial context into the sequence representation without increasing parameter count or attention complexity.

## B    MODEL ARCHITECTURE

### B.1    INPUTS, TOKENIZATION, AND EMBEDDINGS

We ingest multimodal single-cell profiles as a single sequence of length $N \leq 4096$ by concatenating modality-specific tokens (RNA genes, ATAC peaks, ADT proteins; optional spatial tokens). Each token $t$ carries: (i) a *feature identity* embedding $e_{\text{id}}(t)$ (e.g., gene/peak/protein index), and (ii) an *expression-magnitude* encoding $e_{\text{mag}}(t)$ from discretizing counts into $B = 50$ bins. We form per-token inputs by additive composition

$$x_t \;=\; e_{\text{id}}(t) \;+\; e_{\text{mag}}(t) \;+\; e_{\text{pos}}(t),$$

where $e_{\text{pos}}$ is a learned positional encoding. For spatial assays (MERFISH/IMC), we add relative positional encodings over a $k$-NN tissue graph (radius/degree set per dataset) to $e_{\text{pos}}(t)$.

### B.2    ENCODER–DECODER BACKBONE AND POOLING

CELLXPERT uses a Transformer encoder–decoder with sparsely-gated MoE feed-forward blocks in both stacks, layer normalization and residual connections. We pretrain the encoder with masked token prediction (Sec. B.3) and fine-tune a linear classifier on the decoder output for 154-way annotation (Sec. B.4). Unless stated otherwise, we aggregate with *mean sequence pooling*: $h = \frac{1}{N} \sum_{t=1}^{N} z_t$, where $\{z_t\}$ are final-layer token representations. An ablation with a prepended [CLS] shows no benefit over mean sequence pooling (Table 1).

**Figure 6:** Illustration of our sparse MoE block with noisy top-2 routing. We replace the standard dense feed-forward network (FFN) in the Transformer with a two-expert MoE layer. For each input token, a noisy top-2 router independently selects its two highest-scoring experts and returns a gated combination of their outputs. Each expert output multiplied by its router gate value and summed (dotted lines).

### B.3  PRETRAINING OBJECTIVE: MASKED TOKEN MODELING

We randomly replace $p = 0.15$ of tokens with [MASK] and train the encoder to predict the true magnitude bin at masked positions:

$$\mathcal{L}_{\text{MLM}} = -\frac{1}{|M|} \sum_{i \in M} \log p_\theta(x_i \mid x_{\neg M}).$$

Classwise per-bin reconstruction metrics are reported in Figure 9.

### B.4  FINE-TUNING OBJECTIVE: CLASS-WEIGHTED CROSS-ENTROPY

Given pooled representation $h$ and logits $Wh+b$ for $K$ classes, we use inverse-frequency weights

$$\alpha_k = \frac{1/n_k}{\sum_{j=1}^{K}(1/n_j)} \qquad (k = 1, \ldots, K),$$

where $n_k$ is the count of class $k$ in the training set. The weighted loss over a batch $\{(h_b, y_b)\}_{b=1}^{B}$ is

$$\mathcal{L}_{\text{WCE}} = -\frac{\sum_{b=1}^{B} \alpha_{y_b} \log p_\theta(y_b \mid h_b)}{\sum_{b=1}^{B} \alpha_{y_b}}, \quad p_\theta(\cdot \mid h) = \text{softmax}(Wh + b).$$

This mitigates domination by frequent labels and improves macro-F1 on rare cell types (Table 1).

### B.5  SPARSE MOE ROUTING

Each Transformer FFN is replaced by a sparsely-gated Mixture-of-Experts with $E$ experts. Given token embedding $x \in \mathbb{R}^d$, the gate produces noisy logits

$$h = W_{\text{gate}}\, x, \qquad \sigma = \zeta(W_{\text{noise}}\, x), \qquad H(x) = h + \varepsilon \odot \sigma, \ \varepsilon \sim \mathcal{N}(0, I).$$

Here, $d$ is the hidden size; $W_{\text{gate}}, W_{\text{noise}} \in \mathbb{R}^{E \times d}$ are learned gating projections; $\sigma \in \mathbb{R}^E$ is a nonnegative per-expert noise scale (softplus applied elementwise), $\varepsilon \in \mathbb{R}^E$ is i.i.d. Gaussian noise; $I = I_E$ is the $E \times E$ identity.

We keep the top-$k$ entries of $H(x)$ (default $k=2$), set others to $-\infty$, and apply a softmax to obtain sparse mixture weights $g \in \mathbb{R}^E$ (so $\sum_e g_e = 1$). The expert outputs $\{f_e(x)\}_{e=1}^{E}$ are combined as

$$\text{MoE}(x) = \sum_{e=1}^{E} g_e\, f_e(x),$$

where each expert $f_e : \mathbb{R}^d \to \mathbb{R}^d$ is a position-wise FFN. Noisy gating promotes balanced expert utilization and alleviates collapse without additional load-balancing losses. We follow standard token-level dispatch and combine expert outputs within the same sequence shard for efficiency.

### B.6 ATTENTION EFFICIENCY FOR LONG CONTEXT

We use FlashAttention-v2 to compute exact attention with IO-aware tiling, reducing memory traffic and enabling 4k-token contexts at practical memory/throughput (Dao, 2023). FlashAttention-v2's custom `CUDA` kernels require GPUs with compute capability $\geq 8.0$ (Ampere+). On older hardware, CELLXPERT automatically falls back to PyTorch's ATen `scaled_dot_product_attention`, preserving model quality while trading some throughput for compatibility. Kernel selection is resolved at runtime. Checkpoints (`state_dict`) are device-agnostic and unchanged by the choice of kernel, though small numerical differences can appear in activations due to floating-point non-associativity. This dynamic fallback avoids the compatibility problems we observed on legacy GPUs when pipelines assumed FlashAttention-only execution.

### B.7 IMPLEMENTATION NOTES

**Masking schedule.** We sample a fresh Bernoulli mask with rate $p = 0.15$ per batch. Masked positions are predicted only by the encoder's MLM head.
**Sequence packing.** Sequences are truncated to length $\leq 4096$. Spatial encodings are additive.
**Pooling.** Mean pooling is used by default. `[CLS]` pooling is included only for ablation.
**Reproducibility.** All ablations report the same data split and preprocessing.

## C INFRASTRUCTURE, SCALING, AND EFFICIENCY

Pretraining is implemented in `PyTorch 2.5.1` with distributed data parallelism (`DDP`), automatic mixed precision (`AMP`), and `GradScaler`. The workflow comprises two stages trained on the CELLxGENE Discover Census, full details are provided in E.2. FlashAttention-v2 enables up to 16384-token contexts within 32GB, and sparse MoE maintains throughput by raising capacity at near-constant per-token latency.

The `XS` configuration of CELLXPERT is optimized for efficiency and is suitable for scenarios with limited computational resources or for rapid experimentation:

---

**XS Model Configuration (Default)**

**Number of Layers** ($L$): 2
**Number of Attention Heads per Layer** ($H$): 2
**Number of Experts in MoE Layers** ($E$): 4
**Embedding Size** ($d_{\text{model}}$): 128

---

The alternative `M, L, XL` configurations increase the model's capacity to capture more complex patterns in the data, suitable for larger datasets:

---

**M/L/XL Model Configurations**

**Number of Layers** ($L$): 4/8/12
**Number of Attention Heads per Layer** ($H$): 4/8/12
**Number of Experts in MoE Layers** ($E$): 4/8/8
**Embedding Size** ($d_{\text{model}}$): 128/128/128

---

## D PREPROCESSING

We read the raw `h5ad` partitions from CELLxGENE corpus with `anndata` library (Virshup et al., 2021). Each `AnnData` object is filtered to remove cells with fewer than 100 genes and genes present in fewer than 10 cells; tissue and cell-type labels with fewer than 100 cells are also removed. Counts are normalized to 10000 per cell using `scanpy` library (Wolf et al., 2018) and $\log_{10}$-transformed. The top 4096 highly variable genes are selected in each partition. Across training partitions we retain a unified vocabulary of $V = 60{,}530$ genes, including protein-coding, non-coding, lncRNA, small RNA classes (miRNA, snRNA, snoRNA, rRNA, scaRNA) and the full spectrum of pseudogenes.

Per-gene means $\mu_g$ and standard deviations $\sigma_g$ are calculated once on the concatenated training partitions via `Dask` and cached; these parameters are then fixed for both training and test standardization. For any cell $c$ and gene $g$, we compute

$$Z_{c,g} = \frac{X_{c,g} - \mu_g}{\sigma_g}, \tag{1}$$

$$\hat{Z}_{c,g} = \text{clip}\big(Z_{c,g}, -1.96, +1.96\big), \tag{2}$$

where $\text{clip}(x,a,b) = \min(\max(x,a),b)$. Clipping to $[-1.96, 1.96]$ ($\sim 2$ standard deviations correspond to the $2.5^{\text{th}}$ and $97.5^{\text{th}}$ percentiles) bounds extreme values without affecting most data and prevents extreme outliers from destabilizing model training.

### D.1  BINNING-BASED EXPRESSION DISCRETIZATION

We map standardized values to integer bins via quantile binning. Specifically, we flatten all $\hat{Z}_{c,g}$ from pretraining data into a vector $\mathbf{z}$, then split that vector into $B$ equally populated buckets by computing $B + 1$ percentile cut-points (we set $B = 50$ in our implementation).

$$q_k = \text{np.percentile}\big(\mathbf{z}, 100\,k/B\big), \quad k = 0, \ldots, B,$$

caching $\{q_k\}$ for test-time use. Then each $\hat{Z}_{c,g}$ is compared to those stored percentiles $q_k$, whichever interval between adjacent cut-points it falls into determines its integer bin (from 0 up to $B - 1$). An alternative to binning-based discretization is rank-based discretization. While rank-based discretization captures relative expression levels and is robust to batch effects and noise, it discards all magnitude information. In contrast, quantile binning preserves coarse absolute expression levels. Consequently, when downstream tasks depend on expression magnitude such as tasks like differential expression analysis or predicting gene expression changes under perturbations, a binning-based model directly predicts up/down shifts in expression level, whereas a rank-based model might only tell that a gene moves up or down in rank, which is informative but not quantitative. Additionally, to capture subtle magnitude differences for example, identifying rare cell types defined by slight expression changes, binning can detect signals that might be lost in pure rank ordering. Empirically, SCGPT (Cui et al., 2024) and SCBERT (Yang et al., 2022) (bin-based) outperformed GENEFORMER (Theodoris et al., 2023) (rank-based) on imbalanced data with rare cell subpopulations (Alsabbagh et al., 2023). Thus, for tasks like differential expression analysis, perturbation response prediction and rare cell type detection, binning offers an edge.

### D.2  TOKENIZATION

After discretizing expression into $B{=}50$ bins, each cell $c$ is represented by two *aligned sequences* of length $L$: gene IDs $(g_{c,1}, \ldots, g_{c,L})$ with $g_{c,t} \in \{0, \ldots, G\}$ and bin indices $(b_{c,1}, \ldots, b_{c,L})$ with $b_{c,t} \in \{0, \ldots, B-1\}$. We fit a `LabelEncoder` (Pedregosa et al., 2011) on training genes to map each symbol to $[0, V-1]$ and reserve the unused index $V$ for `[CLS]`. If `[CLS]` is enabled, we set $g_{c,1}{=}V$, $b_{c,1}{=}0$ (neutral bin), shift the original pairs by one, and take $L{=}G{+}1$ (otherwise $L{=}G$). We denote the token sequence by $T_c = \big((g_{c,1}, b_{c,1}), \ldots, (g_{c,L}, b_{c,L})\big)$ and identify $N := L \leq 4096$.

### D.3  EMBEDDINGS AND INPUT COMPOSITION

Let $d$ be the embedding dimension (we use $d{=}128$). We define

$$E_{\text{id}} \in \mathbb{R}^{(G+1)\times d}, \quad E_{\text{mag}} \in \mathbb{R}^{B\times d}, \quad P \in \mathbb{R}^{L\times d}.$$

For token position $t \in \{1, \ldots, L\}$,

$$e_{\text{id}}(t) = E_{\text{id}}[\,g_{c,t}\,], \qquad e_{\text{mag}}(t) = E_{\text{mag}}[\,b_{c,t}\,], \qquad e_{\text{pos}}(t) = P[\,t\,].$$

For spatial assays with per-cell coordinates $(x_c, y_c[,z_c])$, we form a per-cell spatial term

$$e_{\text{spatial}}(c) = P^x[\,x_c\,] + P^y[\,y_c\,][\,+P^z[\,z_c\,]],$$

and *tile* it across the sequence. The per-token input is

$$x_{c,t} = e_{\text{id}}(t) + e_{\text{mag}}(t) + e_{\text{pos}}(t)\,[\,+\,e_{\text{spatial}}(c)].$$

## D.4 EXPRESSION ENCODING (SINUSOIDAL, FIXED)

We realize $E_{\text{mag}}$ with a precomputed sinusoidal table (no learned weights). For $b \in \{0, \ldots, B-1\}$ and $i = 0, \ldots, \frac{d}{2} - 1$,

$$E_{\text{mag}}[b, 2i] = \sin\left(b \cdot e^{-\frac{2i}{d} \ln(\text{base})}\right), \qquad E_{\text{mag}}[b, 2i+1] = \cos\left(b \cdot e^{-\frac{2i}{d} \ln(\text{base})}\right),$$

with base=100.

## D.5 VOCABULARY CONSTRUCTION AND DYNAMIC TOKEN ADDITION

We build our gene-token vocabulary once from a `LabelEncoder` mapping each known gene name to a unique integer ID. During fine-tuning, if a gene appears that already has an entry in *gene_to_index*, we reuse its token ID. Otherwise, we allocate the next free ID and append it to both the `LabelEncoder` and the model's embedding matrix. By persisting this updated mapping, CELLXPERT supports incremental vocabulary growth across datasets, enabling *continuous learning* and seamless integration of new gene tokens and modalities during fine-tuning. This mechanism closely parallels ADAPTIVOCAB (Nakash et al., 2025), which pre-allocates embedding slots for tokens absent from the pretraining vocabulary and jointly fine-tunes new and existing embeddings, and resonates with recent work demonstrating that embeddings for previously unseen tokens can be acquired post-hoc by soft token learning (Lester et al., 2021) or prompt tuning (Liu et al., 2023).

## D.6 MAKE MORE WITH LESS DATA

During pretraining we augment dataset size via a novel *bootstrap-style* augmentation technique that replicates each cell with independently permuted gene indices.

> **Bootstrapped Permutation**
>
> Every cell is cloned four times. Each clone receives an independent random permutation of its 4096 highly variable genes before the sequence is truncated to the fixed sequence length of 1024 tokens. This bagging over features approach presents each gene in multiple positional contexts, helps the Transformer learn permutation–invariant patterns, and effectively quadruples the number of training sequences.

Perturb-seq screens often contain less than 100 cells per perturbation. During fine-tuning we therefore boost rare classes with the following heuristics:

> **Data Augmentation Strategies for Perturb-seq Screens**
>
> **Minority oversampling.** With 50% probability a sampled training index is replaced by a random *non-control* cell, equalizing the frequency of perturbed and control classes.
> **Noise injection.** For half the tokens we add a random integer offset in $[-5, 5]$ and clip to $[0, B-1]$, simulating quantization noise around bin boundaries.
> **CutMix.** With 50% probability half of the token positions are replaced by tokens from another cell of the same label to improve robustness to gene dropout.

All operations act on the discretized token sequences and therefore leave the underlying vocabulary, bin edges, and gene statistics unchanged. The combination of bootstrapped permutations and targeted perturbations provides a richer training distribution from the same raw dataset.

# E    PRETRAINING

## E.1    DATASET ACQUISITION AND SPLITTING

We utilized the CELLxGENE Discover Census[1] (version 2023-12-15), accessed via the `cellxgene_census` Python API (Program et al., 2025). To retain unique, non-diseased primary cells, we filtered for:

`organism="Homo sapiens"`: We restrict the dataset to *Homo sapiens* to focus on human cells. This avoids cross-species differences and tailors the analysis to human biology.

`is_primary_data=True`: Because the Census aggregates many studies, the same biological cell can appear in multiple datasets (e.g. in an original study and again in a pooled analysis). We label only one instance as primary to avoid duplicate counting.

`suspension_type` $\neq$ NA: Retains only cells with a defined library preparation: whole-cell (scRNA-seq) or nuclear (snRNA-seq). Ensuring `suspension_type` is specified (i.e. not missing), we remove samples with incomplete metadata.

`disease="normal"`: Only cells from normal (disease-free) tissues or individuals are included, while any cells from diseased or pathological samples (e.g. tumor tissues, disease conditions) are excluded.

This provided a final cohort of 23.88 million single cells from diverse public repositories within the Census. Then, cells were split by `partition_id` into an 80% training set and a 20% test set, with the latter comprising independent experiments to evaluate model generalization.

## E.2    PRETRAINING RECIPE

*Stage 1* trains an encoder-only Transformer as an MLM. Each `h5ad` partition is streamed lazily to the GPUs; the top-$N$ variable genes, drawn from a vocabulary of 60530 transcripts, are quantile-binned into 50 fixed edges. In every mini-batch, every bin token is independently sampled from a Bernoulli($p = 0.15$) distribution and those drawn as 1 are stochastically masked. Cross-entropy loss is computed only on these masked positions. Optimization uses AdamW with $\beta_1 = 0.9$, $\beta_2 = 0.999$ and a weight-decay of $1 \times 10^{-2}$. The learning rate follows a cosine scheduler—1000 warm-up steps to a peak of $1 \times 10^{-3}$ followed by decay to $1 \times 10^{-4}$. AMP and gradient scaling limit memory footprint and preserve numerical stability. Whenever the average masked-loss on a partition decreases, the master process (rank 0) writes a checkpoint containing the full model state dict, optimizer and scheduler states, and AMP scaler state to disk. If training is interrupted or relaunched with the `use_latest_checkpoint` flag, all processes synchronize at startup, load this latest checkpoint, restore epoch/partition counters and hyperparameter schedules exactly as they were, and continue training seamlessly from that point. This ensures no work is lost and providing true fault-tolerant resume capability. Training metrics (masked-token loss, macro precision, recall, $F_1$, learning rate, GPU time, memory) stream live to `Visdom`, and a full bin-confusion matrix, similar to Figure 7, is saved after processing each data partition.

*Stage 2* performs supervised fine-tuning for cell or tissue classification. We freeze the pretrained encoder, feed its activations as the key projections in cross-attention, and attach a decoder of matching depth and width whose output is pooled via either a `[CLS]` token or mean-pooling that can be toggled from the command line. Fine-tuning spans four epochs with AdamW under the same cosine learning-rate schedule. Class imbalance is mitigated through Deferred Re-Weighting (DRW) (Cao et al., 2019): the first two epochs use vanilla cross-entropy, after which inverse frequency class weights are enabled for the remaining two epochs. This two-step schedule enables stable feature representations before biasing the loss toward minority classes. If a checkpoint is resumed with a different loss configuration the optimizer state is reinitialized to avoid resetting momentum. Each cell can be augmented by a user-defined number of random gene-order permutations to improve robustness. Throughout training we log top-1/top-5 accuracy, $F_1$ and confusion matrices to `Visdom`, while UMAP projections are generated only when GPU memory is sufficient.

---

[1]Access to the CELLxGENE dataset is available at: https://cellxgene.cziscience.com/

# F   PRETRAINING RESULTS ON THE CELLxGENE CORPUS

## F.1   EVALUATION ON GENE EXPRESSION RECONSTRUCTION

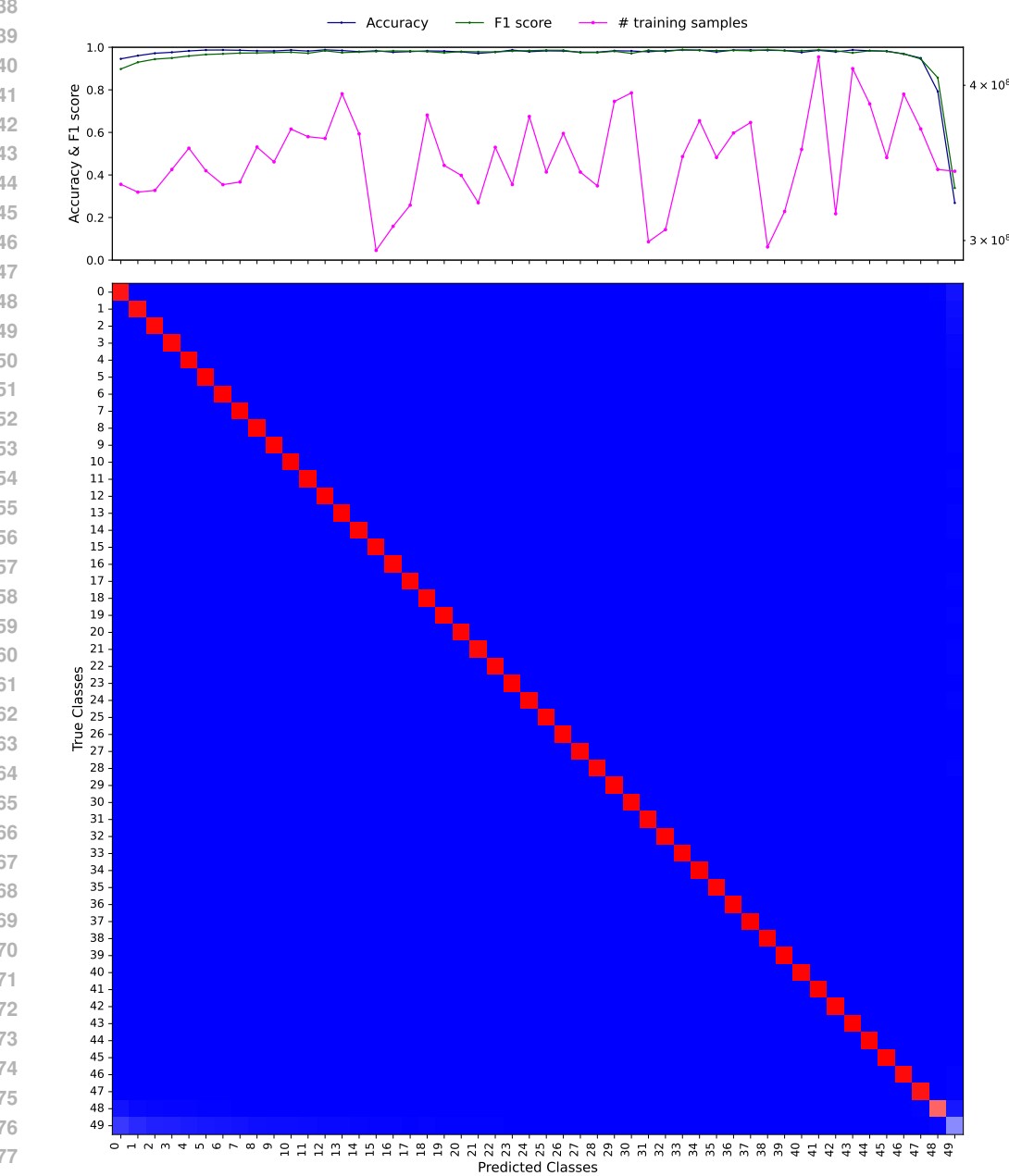

**Figure 7:** Gene expression reconstruction.

## F.2 EVALUATION ON CELL TYPE ANNOTATION

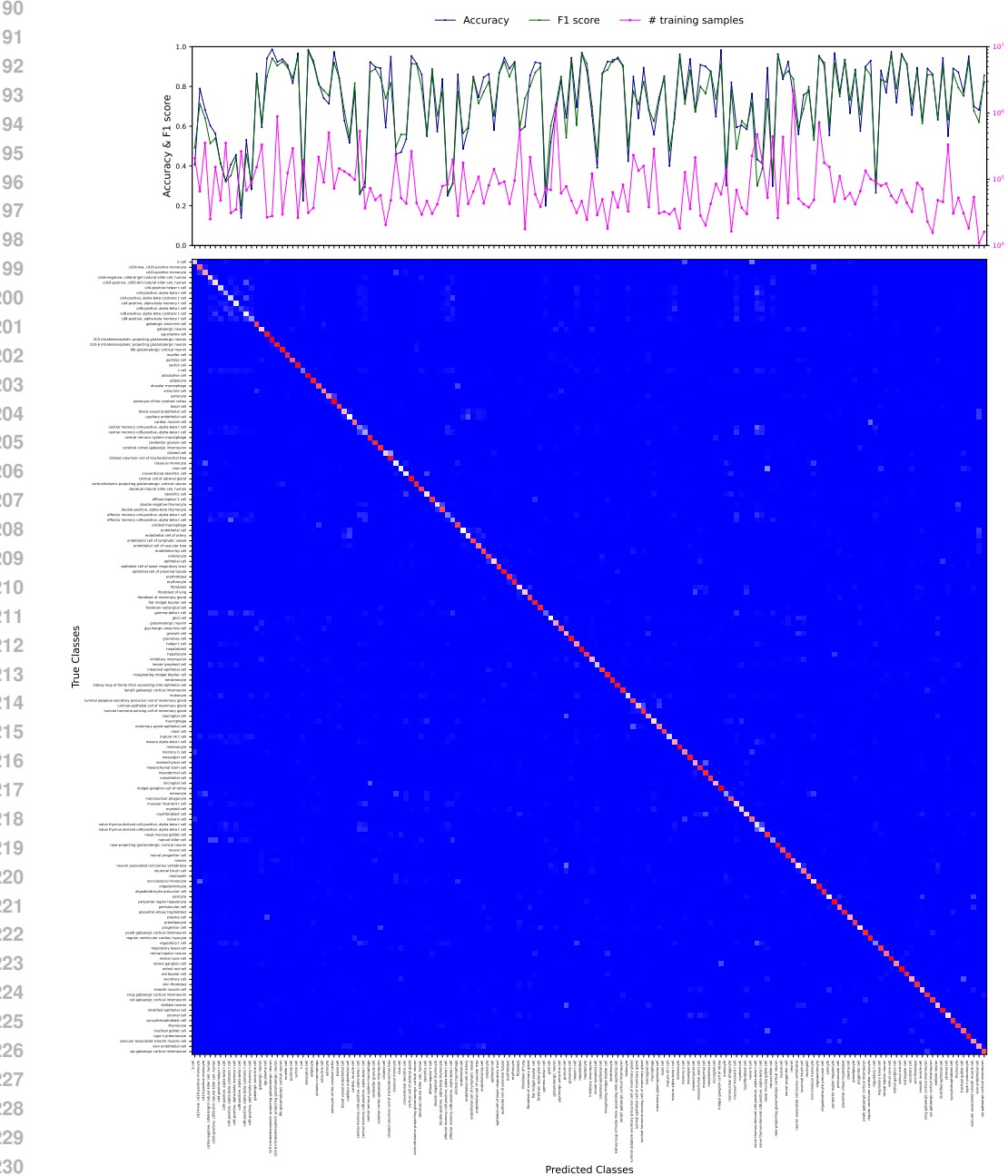

**Figure 8:** Cell type classification.

## F.3 EVALUATION ON TISSUE TYPE CLASSIFICATION

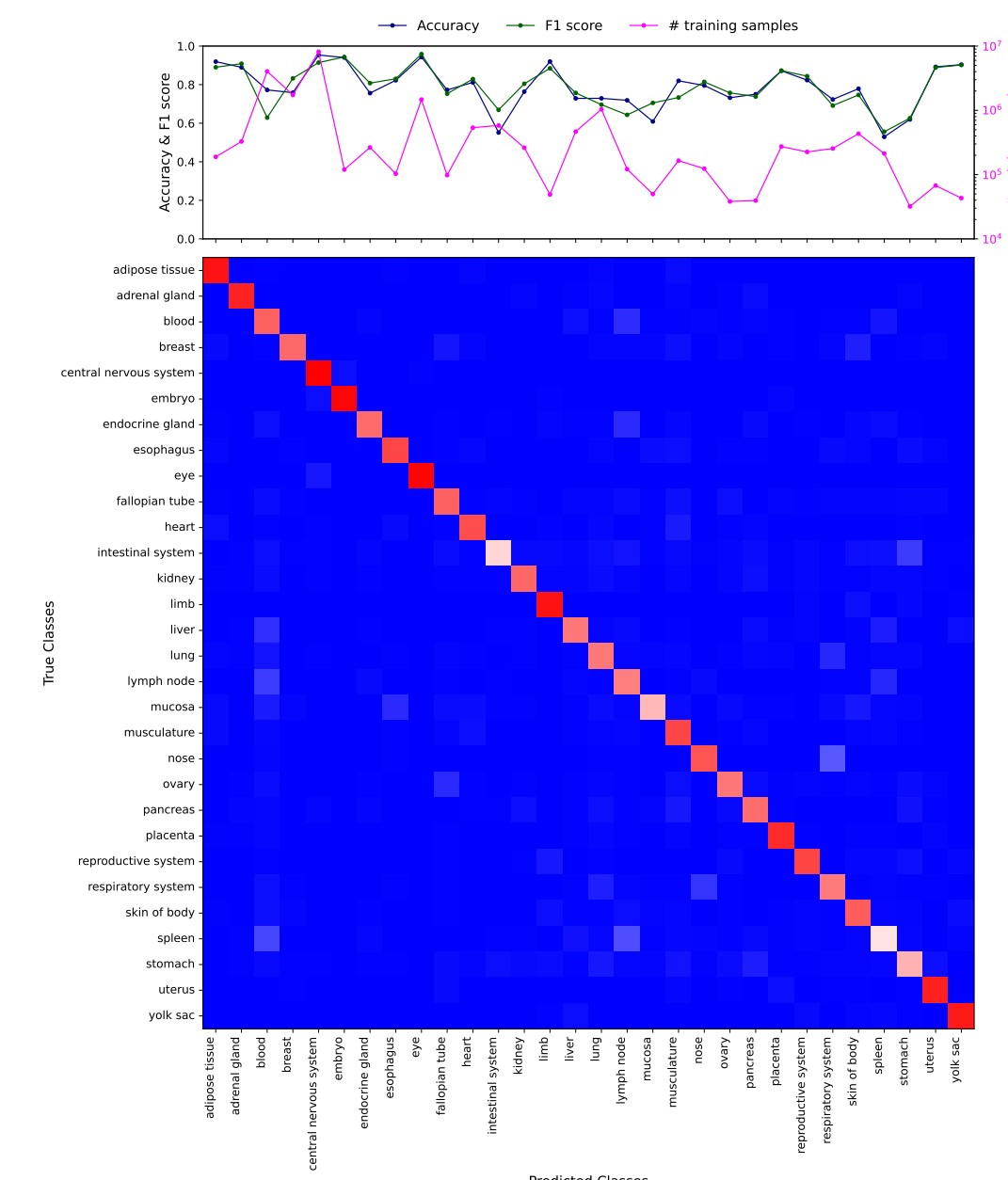

**Figure 9:** Tissue type classification.

## G  FINE-TUNING

### G.1  DECORATOR BASED LoRA ADAPTER

We replace each pretrained linear layer $W \in \mathbb{R}^{d_{\text{out}} \times d_{\text{in}}}$ with a `LoRALinear` wrapper that injects trainable low-rank factors

$$\Delta W = \frac{\alpha}{r} B A, \quad A \in \mathbb{R}^{r \times d_{\text{in}}}, \; B \in \mathbb{R}^{d_{\text{out}} \times r},$$

so that the module's forward pass becomes

$$y = (W + \Delta W) x = W x + \frac{\alpha}{r} B (A x) \; (\textit{Hu et al.}, 2022).$$

The LoRA manager traverses the pretrained model's `nn.Module` tree and, for each attribute in {`queries, keys, values, fc_out`}, replaces the original `nn.Linear` with a `LoRALinear` wrapper. $A$ is initialized via Kaiming-uniform (He et al., 2015) and $B$ is zero so that $\Delta W = 0$ at initialization, preserving pretrained behavior. Freezing $W$ while training only $\{A, B\}$ degraded fine-tuning performance, so we set `freeze_base_model=False` and jointly fine-tune both $W$ and the adapter parameters. All experiments use $r = \alpha = 256$, yielding a scale factor $\alpha/r = 1$ and adding $r(d_{\text{in}} + d_{\text{out}})$ extra parameters per adapted layer.

### G.2  PARKINSON'S DISEASE DATASET

We fine-tuned CELLXPERT model with the (Kamath et al., 2022) dataset. The Kamath et al. dataset comprises gene expression profiling data obtained through high-throughput sequencing. This study focused on midbrain dopamine (DA) neurons in the substantia nigra pars compacta (SNpc), which are critical for voluntary movements, reward processing, and working memory, and are highly susceptible to neurodegeneration in Parkinson's Disease (PD). Utilizing a specialized protocol, DA neuron nuclei from postmortem human SNpc of both PD patients and matched controls were enriched and transcriptionally profiled. The dataset is accessible via GEO accession GSE178265.

## H  ADDITIONAL DETAILS FOR ISP FORMALISM AND BASELINES

### H.1  MEAN-FIELD ONE-SHOT BASELINE

For a perturbation at position $j$, define the masked input $\tilde{\mathbf{x}}^{(j)}$ which contains $x_j = x_j^{(\text{pert})}$ and `[MASK]` at every other index $i \neq j$. The encoder produces per-site logits $\phi_{i,k}(\tilde{\mathbf{x}}^{(j)})$ and posteriors

$$q_i(k \mid \tilde{\mathbf{x}}^{(j)}) \propto \exp\big(\phi_{i,k}(\tilde{\mathbf{x}}^{(j)})\big).$$

The one-shot mean-field estimator factorizes the masked joint as

$$\tilde{p}(\mathbf{x}_{-j} \mid x_j^{(\text{pert})}) \stackrel{\text{MF}}{\approx} \prod_{i \neq j} q_i(x_i \mid \tilde{\mathbf{x}}^{(j)}),$$

ignoring cross-gene dependencies encoded by the model's joint. In practice, this is evaluated under extreme masking ($> 99\%$ masked tokens), which is far outside the pretraining corruption rate ($15\%$) and leads to distribution shift and mode collapse (centroid-like reconstructions) (Cui et al., 2024). Similar collapse can also surface in non-MLM approaches when evaluation emphasizes global shifts (Lotfollahi et al., 2023; Viñas Torné et al., 2025). Under such metrics, linear baselines can match or exceed deep models on novel perturbations (Ahlmann-Eltze et al., 2025; Csendes et al., 2025).

### H.2  ENERGY-BASED VIEW OF CELLXPERT

Let $\mathbf{x} = (x_1, \ldots, x_n)$ be a discrete gene expression vector, $x_i \in \{0, \ldots, B-1\}$.

For masked-gene prediction, an encoder produces logits

$$\phi_i(\mathbf{x}_{-i}) = \big(\phi_{i,0}(\mathbf{x}_{-i}), \ldots, \phi_{i,B-1}(\mathbf{x}_{-i})\big),$$

from which we derive the conditional distribution

$$q_i(k \mid \mathbf{x}_{-i}) \; \propto \; \exp\big(\phi_{i,k}(\mathbf{x}_{-i})\big).$$

Using these logits, we define $n$ positive clique potentials over the full configuration:

$$\psi_i(\mathbf{x}) = \exp\big(\phi_{i,\,x_i}(\mathbf{x}_{-i})\big).$$

This yields the energy function and corresponding Gibbs distribution

$$E(\mathbf{x}) = -\sum_{i=1}^{n} \phi_{i,\,x_i}(\mathbf{x}_{-i}), \qquad p_{\boldsymbol{\theta}}(\mathbf{x}) = \frac{\exp\big(-E(\mathbf{x})\big)}{Z}, \qquad Z = \sum_{\mathbf{x}} \exp\big(-E(\mathbf{x})\big).$$

Equivalently, $p_{\boldsymbol{\theta}}(\mathbf{x}) \propto \prod_{i=1}^{n} \psi_i(\mathbf{x})$. Each $\psi_i$ depends on all coordinates, making the complete graph $K_n$ (a single maximal clique) a suitable undirected graphical model. As the factors are strictly positive, this defines a valid positive Gibbs distribution.

In general, the model's conditional distribution is

$$p_{\boldsymbol{\theta}}(x_i = k \mid \mathbf{x}_{-i}) \; \propto \; \exp\big(\phi_{i,k}(\mathbf{x}_{-i})\big) \times \prod_{j \neq i} \exp\Big(\phi_{j,\,x_j}(\mathbf{x}_{-j}^{(i:=k)})\Big),$$

which does not reduce to $\mathrm{softmax}_k \phi_{i,k}(\mathbf{x}_{-i})$ unless additional compatibility constraints are satisfied. Therefore, training by maximizing $\sum_i \log q_i(x_i \mid \mathbf{x}_{-i})$ serves as a tractable pseudo-likelihood of this energy-based model.

### H.3  Setup for Metropolis–Hastings Sampler

**Notation.** Let $\mathbf{x} \in \{0, \ldots, B-1\}^L$ denote a discretized expression vector over $L$ genes and $B$ bins. Users specify a set of clamped targets $\mathcal{T} \subseteq \{1, \ldots, L\}$ with desired bins $\{x_i^{(\mathrm{pert})} : i \in \mathcal{T}\}$, which are enforced as hard constraints during sampling. Training data are split by class $y \in \{\mathrm{control}, \mathrm{perturbed}\}$ into sets $\mathcal{D}_y = \{\mathbf{x}_y^{(n)}\}_{n=1}^{N_y}$, where control denotes unedited cells and perturbed denotes cells with selective CRISPR-based Perturb-seq edits to specified target(s).

#### H.3.1  Setting Class Anchors via Fréchet Medoids

For each class $y \in \{\mathrm{control}, \mathrm{perturbed}\}$ we summarize the distribution with a small set of in-sample medoids. The single Fréchet medoid is the data point that minimizes the sum of squared Euclidean distances to all other data points in the class,

$$\mathbf{m}_y = \arg\min_{\mathbf{x} \in \mathcal{D}_y} \sum_{\mathbf{x}' \in \mathcal{D}_y} \|\mathbf{x} - \mathbf{x}'\|_2^2.$$

To capture class multimodality, we form a top–$K$ medoid set

$$\mathcal{M}_y \; = \; \{\mathbf{m}_{y,1}, \ldots, \mathbf{m}_{y,K}\} \subset \mathcal{D}_y, \qquad \text{with } \|\mathbf{m}_{y,1} - \boldsymbol{\mu}_y\|_2^2 \leq \cdots \leq \|\mathbf{m}_{y,K} - \boldsymbol{\mu}_y\|_2^2.$$

Unlike the centroid, which can be skewed by outliers, the medoid is an actual observed cell that retains realistic co-expression structure and is empirically more robust (Park & Jun, 2009; Bulté & Sørensen, 2024).

#### H.3.2  log Unnormalized Target Distribution

Let $\mathcal{M}_{\mathrm{perturbed}} = \{\mathbf{m}_k\}_{k=1}^{K}$ be the set of perturbed-class anchors, specifically the top-K medoids from a cluster of cells, selectively perturbed by the same intervention and directed at the same targets. For each gene $i = 1, \ldots, L$, the anchor empirical probability mass function (PMF) is the uniform distribution over the medoid bins: $P_i(b) = \frac{1}{K} \sum_{k=1}^{K} \delta_{m_{k,i}}(b)$ for $b \in \{0, \ldots, B-1\}$. The corresponding cumulative distribution function (CDF) is $F_i^{\mathcal{M}}(b) = \frac{1}{K} \sum_{k=1}^{K} \Vbar[b \geq m_{k,i}]$, which equals $\Pr(X_i \leq b)$ where $X_i$ is uniform over $\{m_{k,i}\}_{k=1}^{K}$.

For a candidate sequence $\mathbf{x}$, the point-mass CDF for gene $i$ is $F_i^{\mathbf{x}}(b) = \mathbb{1}[b \geq x_i]$. The Wasserstein-1 distance between these distributions for each gene simplifies to the L1 norm between CDFs, and across all genes it equals the average L1 distance to the anchors (Vallender, 1974):

$$W_1(\mathbf{x}, \mathcal{M}) = \sum_{i=1}^{L} \sum_{b=0}^{B-1} \left| F_i^{\mathbf{x}}(b) - F_i^{\mathcal{M}}(b) \right| = \frac{1}{K} \sum_{k=1}^{K} \|\mathbf{x} - \mathbf{m}_k\|_1.$$

We precompute, for each gene $g$, the empirical CDF of the perturbed distribution over bins $v \in \{0, \ldots, B-1\}$, denoted as $\mathrm{CDF}_g^{\mathrm{pert}}(v)$. With a single anchor $\mathbf{m}_{\mathrm{pert}}$, this reduces to $\mathrm{CDF}_i^{\mathrm{pert}}(b) = \mathbb{1}[b \geq m_i]$, and the distance is $\sum_i |x_i - m_i|$. The unnormalized log-target distribution for Metropolis-Hastings is then

$$\log \pi(\mathbf{x}) = -\beta \sum_{g=1}^{L} \sum_{v=0}^{B-1} \left| \mathbb{1}[x_g \leq v] - \mathrm{CDF}_g^{\mathrm{pert}}(v) \right|, \quad \beta > 0,$$

subject to the hard constraint $\mathbf{x}_{\mathcal{T}} = \mathbf{x}_{\mathcal{T}}^{(\mathrm{pert})}$ on genes selectively targeted during the perturbation. This facilitates the transport of the control cell state to the perturbed distribution by minimizing the optimal transport cost while enforcing fixed values. The multi-anchor case is a special instance where $\mathrm{CDF}_g^{\mathrm{pert}}(v) = F_g^{\mathcal{M}}(v)$, and the factor $1/K$ can be absorbed into $\beta$.

This formulation evaluates in $O(B\,L)$ time per candidate, since the Wasserstein-1 cost reduces to a sum of 1D marginal costs via precomputed CDF differences. In contrast, general optimal transport (OT) solvers, such as the Hungarian algorithm ($O(N^3)$) or Sinkhorn algorithm ($O(N^2/\epsilon)$ (Cuturi, 2013)), scale poorly with dimension and support size $N$, making our approach suitable for high dimensional gene sequences in large Perturb-seq datasets.

### H.4 TEMPERATURE ABLATION FOR METROPOLIS–HASTINGS PROPOSALS

To characterize the effect of the proposal temperature $\tau$ on ISP response prediction, we perform an ablation over $\tau$ while keeping the model, training procedure, and MCMC budget fixed. We sweep $\tau \in \{0.1, 0.5, 1.0, 2.0, 4.0, 5.0, 8.0, 10.0\}$ and evaluate ISP performance on the Replogle K562 benchmark. For each value of $\tau$ we report the mean Pearson-$\Delta_{\mathrm{ctrl}}$ under the standard control-referenced definition and the mean Pearson-$\Delta_{\mathrm{pert}}$ under the SYSTEMA perturbed-reference definition.

Figure 10 shows that both metrics improve monotonically as $\tau$ increases from 0.1 to roughly 2.0 and then enter a broad plateau. The stricter SYSTEMA Pearson-$\Delta_{\mathrm{pert}}$ follows the same pattern. ISP performance is not hypersensitive to the exact choice of $\tau$ it is in a reasonable range. We set $\tau = 2$ in the main experiments as a point in this stable regime that balances exploration and exploitation while avoiding the degradation observed at very low temperatures.

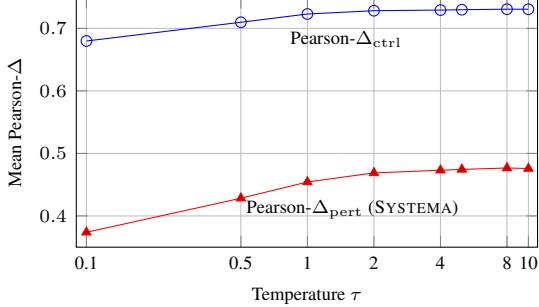

**Figure 10:** Sensitivity of Metropolis–Hastings ISP performance on Replogle K562 to the proposal temperature $\tau$.

## H.5 MCMC CONVERGENCE AND ACCEPTANCE DYNAMICS

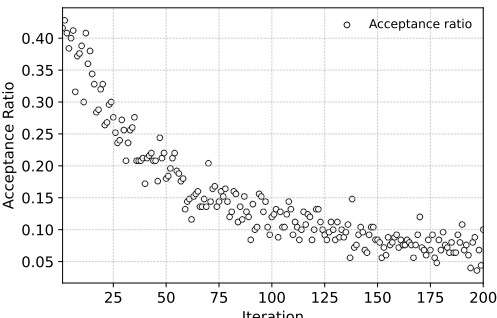

**Figure 11:** Metropolis–Hastings acceptance ratio $[0, 1]$ per iteration during ISP sampling for a PMF1 knockdown. The initial high acceptance rates indicate fast mixing and at later iterations proposals differ only slightly and are accepted more selectively, indicating convergence of the Markov chain.

## H.6 MCMC COMPUTE BUDGET AND PRACTICAL RUNTIME

Our MCMC sampler runs purely at inference time. For each batch of control cells we iterate a blockwise Metropolis–Hastings chain without gradients, so the computation consists only of encoder forward passes. At iteration $t$ we mask a random block $M_t$ of non-target genes and evaluate two MLM forward passes (one on $\mathbf{x}_{-M_t}$ and one on $\mathbf{x}'_{-M_t}$), which makes the per-step cost linear in the batch size and sequence length and linear in the number of MCMC steps.

On a single NVIDIA H100 NVL, our blockwise MH sampler runs at 8 iterations per second, requiring 25 seconds per chain with 200 iterations for a batch of 256 control cells. Covering the full high-confidence Replogle subset (with 290 perturbation targets, each requiring 10 to 15 such batches) therefore takes approximately 24 hours of compute. Since ISP sampling is purely inference-time, this overhead is modest relative to pretraining or fine-tuning, and the total cost scales linearly with the number of control cells per line (about 2.5k in Replogle, processed in batches), the context length (up to 4096 tokens), and the number of MH steps $T$.

## I SPATIAL DATA INTEGRATION

### I.1 SPATIAL TRANSCRIPTOMICS (MERFISH)

The MERFISH platform provides single-molecule sensitivity (Chen et al., 2015) and subcellular spatial resolution (Moffitt et al., 2018), allowing reconstruction of a high-resolution 3D point cloud of cells and their microenvironments. Its primary limitation is that only a predefined gene set is measured, rather than the full transcriptome. For our analysis, we use a subset of the MERFISH dataset comprising 73655 cells from (Moffitt et al., 2018), where each cell includes precise 3D spatial coordinates $(x, y, z)$ in micrometers, a 161-dimensional feature vector based on MERFISH spot counts, and one of 16 cell type annotations: Ambiguous, Astrocyte Endothelial 1-3, Ependymal, Excitatory Inhibitory Microglia, OD Immature 1-2 OD Mature 1-4, and Pericytes. The 161 features consist of 155 MERFISH gene targets (comprising 85 known markers curated from prior single-cell and bulk RNA studies plus 70 novel markers identified by differential expression analysis) along with 6 control features (5 blank barcodes for background noise estimation and *cFos* for detecting recently activated neurons).

### I.2 SPATIAL PROTEOMICS (IMAGING MASS CYTOMETRY)

IMC uses metal isotope tagged antibodies to label proteins in tissue slices, followed by laser ablation and time-of-flight mass spectrometry to detect these tags (Giesen et al., 2014). This generates high-dimensional images, where each pixel measures dozens of protein markers at subcellular resolution. In our analysis, we use a subset of the IMC breast cancer dataset, which comprises 720 high-dimensional images from 352 patients and yields approximately $1.7 \times 10^6$ segmented cells (Jackson et al., 2020).

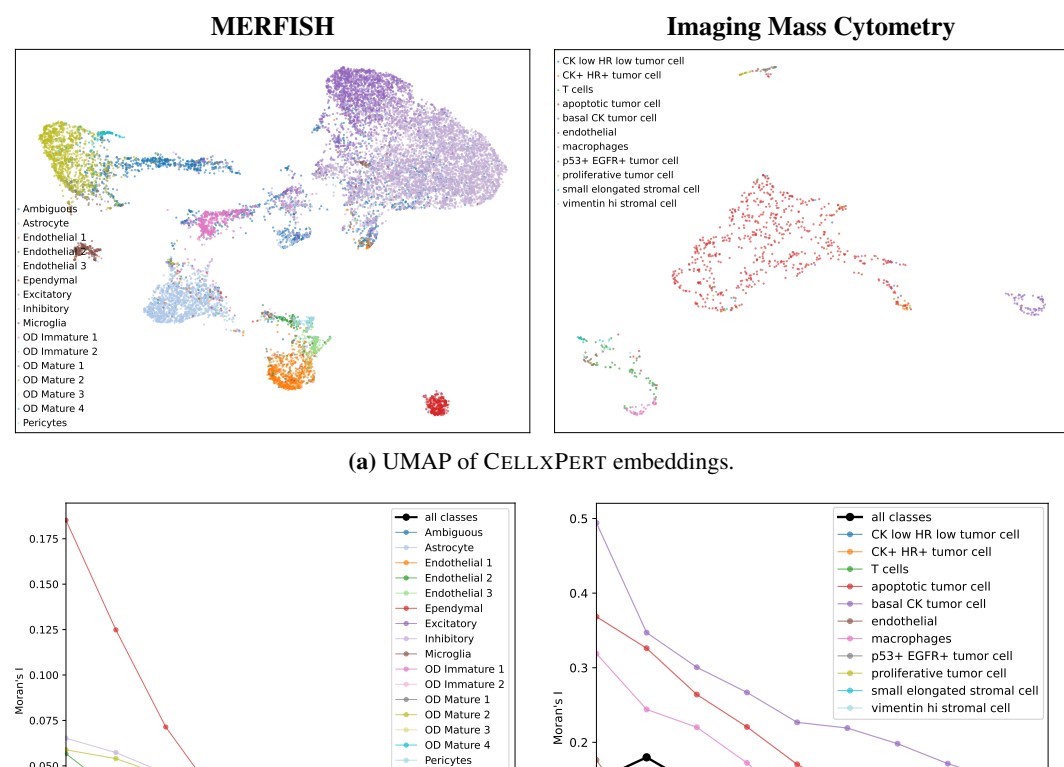

**(a)** UMAP of CELLXPERT embeddings.

**(b)** Variogram analysis.

**Figure 12:** UMAP embeddings and variogram analysis produced by CELLXPERT. The left panel shows MERFISH and the right panel shows IMC. Points are colored by predicted cell type to visualize clustering in the embedding space. Variograms compute Moran's $I$ over increasing radii from predicted labels to quantify spatial organization. Positive $I$ indicates clustering, values near zero indicate randomness, and negative values indicate dispersion (Moran, 1950). In MERFISH the ependymal types decay rapidly which indicates localized niches, whereas vascular and mature oligodendrocyte states retain positive $I$ over larger distances which is consistent with perivascular corridors and lineage domains. In IMC the epithelial tumor classes start with high $I$ and decay slowly which indicates compact tumor islands, while T cells and stromal classes show low and rapidly decaying $I$ which is consistent with their dispersed distributions.

For each cell, the data includes 2D spatial coordinates $(x, y)$ in micrometers, intensity values across 35 protein channels, and a cell-type label derived from PhenoGraph clustering (Levine et al., 2015). The cell types are: CK low HR low tumor cell, CK+ HR+ tumor cell, T cells, apoptotic tumor cell, basal CK tumor cell, endothelial, macrophages, p53+ EGFR+ tumor cell, proliferative tumor cell, small elongated stromal cell, and vimentin hi stromal cell. The 35-channel panel includes clinical markers (ER, PR, HER2), proliferation markers (Ki-67), lineage markers (PanCK, Vimentin, CD45, CD3, CD8, CD68, CD20, CD31, $\alpha$SMA, Fibronectin), and other proteins. Overall, the IMC panel targets a mix of cell surface, intracellular, and extracellular matrix proteins to enable detailed phenotyping of tumor, immune, and stromal cells in breast cancer tissues.

### I.3 EXPERIMENTAL RESULTS

**MERFISH** The UMAP of CELLXPERT embeddings in Figure 12a shows clear clusters for oligo-dendrocyte lineages, ependymal cells, astrocytes, and excitatory and inhibitory neurons. Spatially

compact and molecularly distinctive types achieve high per class $F_1$ scores. *Ependymal* reaches 91% with $n$=395. *Astrocyte* reaches 88% with $n$=1558. *Endothelial 1* reaches 87% with $n$=788. *Inhibitory* reaches 84% with $n$=5015. *Excitatory* reaches 78% with $n$=2402. Weighted precision is 80%. Weighted recall is 76%. Weighted $F_1$ is 77%. A total of 52.7% of cells belong to classes with macro-$F_1$ at least 80%. Errors concentrate in rare or closely related oligodendrocyte subtypes. *OD Mature 4* has $F_1$ of 28% with $n$=81 and exhibits high recall with low precision, which suggests over assignment from neighboring states. *OD Mature 2* has $F_1$ of 71% with precision of 92% and recall of 59% with $n$=1118 and shows the opposite pattern, which indicates conservative decision boundaries. The *Ambiguous* class has $F_1$ of 52% with $n$=1855 and captures heterogeneous cells as expected.

Neighborhood enrichment computed on CELLXPERT predictions for the MERFISH dataset in Figure 4 reveals biologically coherent tissue modules. A neurovascular unit emerges in which *Endothelial 2* is strongly enriched with *Pericytes* and positively associated with *Astrocytes*. This pattern reflects the endothelial to pericyte to astrocyte triad known to support blood brain barrier function, perfusion, and homeostasis (Iadecola, 2017; Presa et al., 2020). *Microglia* show weaker secondary co-occurrence with this vascular compartment. *Ependymal* cells are strongly self enriched. Oligodendrocytes separate into immature and mature sub blocks with moderate cross links, which is consistent with a stage structured lineage continuum. *Excitatory* and *Inhibitory* neurons are mutually enriched and show their strongest depletions against the oligodendrocyte block, especially mature states. As shown in Figure 12b, the variogram for all classes decays toward zero with increasing radius, which reflects a transition from local microdomains to tissue scale heterogeneity. Class specific curves highlight niche scales. Ependymal types decay fastest, which indicates short range structures. Vascular and several oligodendrocyte states sustain positive $I$ over larger radii, which is consistent with perivascular corridors (Maki et al., 2015).

**IMC** CELLXPERT attains strong per class $F_1$ on the IMC dataset. *Macrophage* reaches 87% with $n$=40. *Apoptotic Tumor Cell* reaches 84% with $n$=635. *Basal CK Tumor Cell* reaches 80% with $n$=76. *Endothelial* reaches 74% with $n$=16. *T Cell* is moderate at 60% with $n$=63. The UMAP in Figure 12a recovers tumor epithelial subtypes such as *Basal CK Tumor Cell* and *CK+ HR+ Tumor Cell*, immune cells including *T Cell* and *Macrophage*, stromal cells such as *Vimentin High Stromal Cell* and *Small Elongated Stromal Cell*, and functional states such as *Proliferative Tumor Cell* and *Apoptotic Tumor Cell*. Class imbalance is substantial because apoptotic tumor cells dominate. The macro-$F_1$ is 60%. The weighted $F_1$ is 78% with weighted precision of 85% and weighted recall of 75%. Lower $F_1$ values occur in sparse or phenotypically overlapping classes. *p53+ EGFR+ Tumor Cell* has 41% with $n$=31. *Proliferative Tumor Cell* has 48% with $n$=22. *Small Elongated Stromal Cell* has 42% with $n$=17. These patterns are consistent with diffuse and transitional morphology in the embedding space.

Neighborhood enrichment based on CELLXPERT predictions groups classes into coherent tissue modules in Figure 4. Epithelial tumor classes *Basal CK* and *CK+ HR+* show strong self enrichment and only moderate mutual adjacency, which is consistent with contiguous tumor patches rather than one fused epithelial block. *Macrophage* co-enriches with *T Cell* and with *Vimentin High Stromal Cell*, which marks an immune stromal interface common in solid tumors (Wu et al., 2021; Jackson et al., 2020). *Endothelial* shows modest co-enrichment with stromal and immune compartments, which is consistent with vascular tracks that outline tumor nests (Binnewies et al., 2018). In contrast, *Macrophage* is depleted near the apoptotic tumor state and apoptotic tumor is depleted relative to major stromal classes. We compute Moran's $I$ across increasing radii to obtain variograms of spatial coherence. In IMC, epithelial classes start with high $I$ at small radii and decay slowly, which indicates compact tumor islands that persist over larger scales. *T Cell* and stromal classes exhibit low and rapidly decaying $I$, which aligns with their dispersed distributions.

## J  MULTI-OMIC DATA INTEGRATION

We use the OpenProblems NeurIPS 2021 bone-marrow mononuclear cell (BMMC) benchmark hosted under GEO accession GSE194122.[2] This resource provides matched CITE-seq, which pairs RNA with antibody-derived tags, and 10x Multiome, which pairs RNA with ATAC-seq. Because both assays include an RNA layer, we apply the same preprocessing pipeline to RNA across the

---

[2] https://www.ncbi.nlm.nih.gov/geo/query/acc.cgi?acc=GSE194122

two datasets. Specifically, we keep cells that express at least 200 genes, remove genes detected in fewer than 10 cells, and exclude cells whose mitochondrial RNA fraction exceeds 15%. We then normalize each cell to a total of 10,000 counts and apply a log-transform. Statistically variable genes are retained for modeling. Before training, features are standardized using the mean and standard deviation estimated on the training set, with values clipped to a reasonable range to limit outliers. Finally, we convert continuous features into discrete tokens using quantile binning. By default we use 50 quantile bins whose edges are fit on the training data and then applied unchanged to the test data. The quality-control and normalization steps follow standard Scanpy practice.[3]

### J.1 ATAC-seq Data Preprocessing

For the ATAC layer we use modality-aware quality control derived from common Multiome metadata. We retain cells that have at least 1000 unique nuclear ATAC fragments, that concentrate at least 30% of reads within called peaks, and that show a nucleosome signal no greater than 2.0. We preserve an unmodified copy of the raw peak counts and compute a normalized, log-transformed peak matrix for analysis. We then select the most variable peaks and intersect the RNA and ATAC pass lists so that only cells of acceptable quality in both modalities are kept. The resulting gene and peak features are standardized with the training set statistics and discretized using the same scheme as the RNA layer. We form fixed length token sequences that mix genes and peaks as inputs to the model.

### J.2 CITE-seq Data Preprocessing

For the ADT layer we require adequate protein signal per cell and reasonable prevalence per protein. Specifically, we keep cells with at least 100 total ADT counts and at least five proteins detected, and we retain proteins that are expressed in at least five cells. We preserve raw ADT counts and then apply per-cell centered log-ratio (CLR) normalization to obtain a scale that is robust to depth differences across cells. After intersecting the RNA and ADT pass lists, we standardize protein features with training-set statistics and discretize them using the same approach as the RNA layer with global quantile binning at 50 bins. We then construct fixed length token sequences that mix genes and proteins. The train–test splitting strategy mirrors the ATAC setup.

### J.3 Experimental Results

**RNA + ATAC (10x Multiome).** We evaluated CELLXPERT on matched RNA and ATAC token budgets in BMMC and found that gene expression is the primary driver of accuracy while chromatin accessibility provides only modest complementary signal. Test-1 accuracies presented in Figure 5 corroborates this conclusion. With zero peaks, accuracy increases from 75.3% at 512 genes to 80.1% at 1024 genes and 81.5% at 2048 genes. Peaks alone are weak with 35.4%, 43.9%, and 54.0% at 512, 1024, and 2048 peaks. Adding peaks on top of genes yields small gains that diminish as the gene budget grows. At 512 genes, introducing 2048 peaks improves accuracy from 75.3% to 78.0% which is a gain of 2.7%. At 1024 genes, the best mix reaches 80.7% which is an improvement of 0.6%. At 2048 genes, the best mix is 82.4% with 512 peaks which is a 0.9 point gain, whereas 2048 peaks slightly reduce accuracy to 81.2%. Per-class $F_1$ analysis aligns with these patterns. The median per-class $F_1$ favors RNA at 81.2% compared with 49.1% for ATAC. RNA exceeds ATAC on every class with the largest improvements in antibody secreting and myeloid populations where *Plasma Cell* reaches 83.5% vs. 24.6% which is a gain of 58.9%, *CD16$^+$ Monocytes* reach 90.4% vs. 41.0% which is a gain of 49.5%, *cDC2* reaches 77.9% vs. 35.6% which is a gain of 42.2%, and *ID2 High Myeloid Progenitors* reach 48.5% vs. 8.3% which is a gain of 40.2%. RNA also leads for *Natural Killer* and *Naive B Cells* with 87.8% vs. 50.8% which is a gain of 37.1% and 88.2% vs. 60.1% which is a gain of 28.1%. ATAC retains lineage level signal for broad programs such as *CD8$^+$ T* at 69.2%, *Erythroblast* at 75.6%, and *Transitional B* at 60.8% yet it under resolves activation and early progenitor states, for example *CD4$^+$ T Activated* at 59.6% for RNA vs. 37.9% for ATAC and *Granulocyte Or Monocyte Progenitors* at 62.2% vs. 31.5%. Biologically, these results are consistent with RNA capturing immediate effector and activation programs while peak level accessibility at this granularity reflects broader lineage permissivity and lacks the discriminative detail needed for closely related or transient states.

---

[3] https://scanpy.readthedocs.io/en/stable/generated/scanpy.pp.calculate_qc_metrics.html

**RNA + CITE-seq (ADT).** Across the CITE-seq benchmark, the mixed setting with both modalities and a larger token budget (200 proteins + 3896 genes; 4096 tokens total) delivered the strongest aggregate performance. As shown in Figure 5, Test-1 accuracy increased from 79.2% with RNA-only (3896 genes) and 77.5% with protein-only (200 ADTs) to 85.7% with the mixed model, gains of +6.5% and +8.2%, respectively. At the class level (reported with $F_1$ for interpretability), the mixed model achieved the highest $F_1$ on 23 of 45 cell types, covering 62.6% of test cells, while RNA-only and ADT-only were best on 10 (19.9%) and 12 (17.6%) classes, respectively. Proteins were decisive for activation and receptor-defined T/NK phenotypes: for *CD4$^+$ T Activated*, $F_1$ jumped from 45.6% (RNA) to 83.3% (ADT) and 85.3% (mixed), and for *CD8$^+$ T CD69$^+$ CD45RA$^+$* it increased from 29.7% (RNA) to 69.2% (ADT) and 64.8% (mixed). KIR-stratified NK subsets also favored proteins (*NK CD158e1$^+$*: 50.1% RNA $\rightarrow$ 76.3% ADT $\rightarrow$ 71.4% mixed). Conversely, transcriptionally stereotyped erythroid and plasmacytoid dendritic cells were gene-driven (*Reticulocyte*: 98.4% RNA vs. 77.5% ADT vs. 97.4% mixed; *pDC*: 97.5% RNA vs. 94.8% ADT vs. 96.6% mixed). Crucially, combining modalities rescued several hard classes where a single modality faltered: *Plasmablast IGKC$^+$* improved from 29.9% (RNA) and 3.6% (ADT) to 62.7% (mixed); *Plasma Cell IGKC$^-$* from 37.6% and 31.1% to 58.7%; *G/M Progenitors* from 67.6% and 63.1% to 75.2%; and *NK* from 74.4% and 63.9% to 81.2%. The mixed model also corrected precision–recall imbalances introduced by protein-only classification in rare innate populations (*ILC* precision rose from 6.0% to 30.2% while recall remained high at 93.6%, lifting $F_1$ from 11.0% to 45.7%). Taken together, adding the protein modality and expanding the token budget improved overall Test-1 accuracy and yielded large, class-specific gains for receptor/activation-defined states, while preserving near ceiling performance for gene-dominated lineages.

## K  BROADER IMPACT

We aim to develop a rank based target recommendation system that proposes gene perturbations capable of driving a cell toward a desired state, which supports therapeutic target identification. Given a disease state and a reference state, our ISP sampler generates counterfactual expression for single gene perturbations and small combinations, then ranks candidates by the predicted shift toward the reference. We pair these rankings with attributions from attention weights and gradients for gene level saliency maps and pathway level summaries computed with gene set enrichment analysis (GSEA) (Chefer et al., 2021a;b; Selvaraju et al., 2017; Smilkov et al., 2017; Srinivas & Fleuret, 2019; Sundararajan et al., 2017; Bach et al., 2015; Lundberg & Lee, 2017; Nam et al., 2020; Shrikumar et al., 2017). Shortlists are filtered by druggability, essentiality screens, and off target risk. This workflow could reduce costs and shorten timelines for target discovery in oncology and immunology. Risks primarily stem from nuisance factors and batch effects across datasets and cohorts. We mitigate these risks with rigorous cross-validation, including stratified and leave-one-cohort-out splits, and we assess out-of-distribution generalization on held-out cohorts, labs, and platforms.

## L  RELATED WORK

Single-cell sequencing is prone to technical artifacts, dropout events, and batch effects (Hicks et al., 2018; Stuart et al., 2019), which are further amplified in weakly supervised settings by noisy cell labels. Even the largest single-cell datasets to date, the Tahoe-100M dataset (Zhang et al., 2025) of 100 million transcriptomic profiles from 50 cancer cell lines exposed to 1100 small molecule perturbations and the CZ CELLxGENE data corpus (Program et al., 2025) a collection of 93 million cells (63% of them are from human), remain constrained by these confounders. Moreover, scaling laws for Large Language Model (LLM) pretraining predict diminishing returns from enlarging model size alone (Hoffmann et al., 2022; Kaplan et al., 2020); hence, synthetic data generation via generative models, in silico perturbations, and data augmentation strategies is essential both to expand and diversify training examples in line with scaling laws and to confer adversarial robustness against nuisance factors (Nouri, 2025).

The primary bottleneck in single-cell foundation modeling is thus the availability of high-quality observational data (vast, diverse cell atlases) and especially interventional data (pooled CRISPR-based perturbation screens linking cause to effect), not model capacity. Large scale models such as TranscriptFormer (Pearce et al., 2025) (542M params.), Teddy (Chevalier et al., 2025) (400M params.) and Geneformer (Theodoris et al., 2023) (106M params) exhibit similar performance ceilings when

trained on existing datasets. Recent research (Rood et al., 2024) proposes complementing the Human Cell Atlas with a Perturbation Cell Atlas to enable *truly causal foundation models*. In this landscape, CELLXPERT, despite its modest 26.3 M parameters, achieves a very competitive accuracy by integrating multimodal single-cell signals in an efficient sparse Mixture-of-Experts Transformer, demonstrating that data diversity and quality outweigh sheer model size.

CELLXPERT instantiates ISP as constrained generation in discrete sequence space. We build on CGMH, the Markov random field view of BERT, and Gibbs-style sampling for sequence design (Miao et al., 2019; Wang & Cho, 2019; Johnson et al., 2021). Concretely, we propose MLM edits of gene tokens and accept or reject them with a Metropolis–Hastings rule that steers trajectories toward class anchors estimated from control and CRISPR-perturbed cells. The result is an on-manifold, iterative procedure that replaces hard rank edits and one-shot imputations, stays close to the pretraining distribution through low-rate masking, and mitigates mean regression while preserving biologically meaningful variability.

**Table 8:** Overview of single-cell foundation models and supported capabilities.

**(a)** Comparison of single-cell foundation models used in this study. The table reports architecture, parameter count, vocabulary or gene list size, maximum input sequence length in tokens or genes, and pretraining data scale. Values are taken from the original papers or public model repositories.

| Model | Design | Num. Params. | Vocab. Size | Max. Input Seq. | Pretraining Data |
|---|---|---|---|---|---|
| **CELLXPERT**[*] | Encoder–decoder transformer; MoE[1] | 26.3M | 100K tokens | 16384 tokens | 23.6M scRNA-seq cells |
| scGPT[*] | Decoder-only transformer | 41.9M | ~30K genes | 1200 tokens | 33M scRNA-seq cells |
| GENEFORMER[*] | Decoder-only transformer | 10.2M; 106M[3] | 25429 genes | 4096 genes | 104M scRNA-seq cells |
| GEARS | Encoder–decoder transformer | ~100M | 19264 genes | N/A | 50M scRNA-seq cells |
| CELLPLM[*] | Gaussian mixture variational encoder | 82.4M | N/A | 1000 genes | 9M scRNA-seq cells |
| scBERT[*] | Decoder-only transformer | 8.4M | ~20K genes | 16000 genes | 1.2M scRNA-seq cells |
| xTRIMOGENE[*] | Encoder–decoder transformer | ~100M | 19264 genes | N/A | 50M scRNA-seq cells |

[*] Models using Masked Language Modeling (MLM).
[1] MoE: Mixture of Experts.
[2] SRT: Spatially Resolved Transcriptome Data.
[3] Geneformer 106M: Available through Nvidia's BioNeMo Framework.

**(b)** Feature comparison across single-cell foundation models evaluated in this work. Each column indicates whether the released model and its public repositories provide the stated capability. A check mark indicates support. A cross indicates not supported or not reported.

| Model | Multi-Omic Data Integration | Cell Type Annotation | Gene Function Prediction | Perturbation Prediction | Modeling Spatial Omics |
|---|---|---|---|---|---|
| **CELLXPERT** | ✓ | ✓ | ✓ | ✓ | ✓ |
| scGPT | ✓ | ✓ | ✓ | ✓ | ✓ |
| GENEFORMER | ✗ | ✓ | ✓ | ✓ | ✗ |
| CELLPLM | ✗ | ✓ | ✗ | ✓ | ✓ |
| scBERT | ✗ | ✓ | ✗ | ✗ | ✗ |
| xTRIMOGENE | ✗ | ✓ | ✗ | ✓ | ✗ |

# M CELLxGENE: DETAILED PERFORMANCE RESULTS FOR CELL TYPE ANNOTATION

We assess whether pretraining yields discriminative structure at single-cell resolution across a broad ontology of 154 largely overlapping identities. CELLxPERT attains high $F_1$ on abundant, transcriptionally stereotyped lineages, including *hepatoblasts* (n=46,665; $F_1$=96.9%), *retinal rod cells* (n=68,690; $F_1$=95.8%), *oligodendrocytes* (n=721,082; $F_1$=95.0%), *type II pneumocytes* (n=18,172; $F_1$=92.7%), and *cortical cells of adrenal gland* (n=263,029; $F_1$=91.6%). These results align with lineage-specific marker programs (e.g., AFP/CYP3A7/DLK1 for hepatoblasts; RHO/GNAT1 for rods; MBP/MOG/PLP1 for oligodendrocytes; SFTPC/ABCA3 for AT2 cells; CYP11B2/STAR for adrenal cortex), which produce well-separated decision boundaries. Errors concentrate among closely related or low-abundance subtypes that share gradient or transitional markers, notably along the CD14/CD16 monocyte continuum, NK maturation ($CD56^{bright}$/$CD16^+$), and CD4/CD8 $\alpha\beta$ T-cell states, consistent with known biological overlaps. To make evaluation robust to long-tail classes, we emphasize macro-$F_1$ in Table 9 and provide confusions along with per-class support (#cells) in Figure 8, enabling a calibrated interpretation of performance under class imbalance.

**Table 9:** Classification performance for 154 cell types

| Cell Type | Accuracy (%) | $F_1$ (%) |
|---|---|---|
| absorptive cell | 98.44 | 97.57 |
| hepatoblast | 96.97 | 96.94 |
| near-projecting glutamatergic cortical neuron | 96.23 | 96.28 |
| retinal rod cell | 96.39 | 95.80 |
| retinal cone cell | 97.37 | 95.79 |
| sertoli cell | 96.66 | 95.05 |
| oligodendrocyte | 95.46 | 95.04 |
| melanocyte | 96.11 | 94.68 |
| l2/3 intratelencephalic projecting glutamatergic neuron | 98.69 | 94.20 |
| kidney loop of henle thick ascending limb epithelial cell | 94.57 | 93.90 |
| keratinocyte | 92.32 | 93.46 |
| placental villous trophoblast | 93.53 | 93.33 |
| granulosa cell | 94.42 | 92.81 |
| type ii pneumocyte | 95.18 | 92.65 |
| epithelial cell of proximal tubule | 94.46 | 92.48 |
| l6b glutamatergic cortical neuron | 93.72 | 92.43 |
| astrocyte of the cerebral cortex | 97.40 | 92.06 |
| oligodendrocyte precursor cell | 90.67 | 92.00 |
| adipocyte | 92.81 | 91.87 |
| cortical cell of adrenal gland | 95.34 | 91.63 |
| erythrocyte | 92.36 | 91.58 |
| periportal region hepatocyte | 96.73 | 91.53 |
| hepatocyte | 88.11 | 91.51 |
| midget ganglion cell of retina | 98.30 | 91.31 |
| corticothalamic-projecting glutamatergic cortical neuron | 91.50 | 91.18 |
| rod bipolar cell | 91.28 | 91.17 |
| l2/3-6 intratelencephalic projecting glutamatergic neuron | 92.37 | 90.45 |
| lamp5 gabaergic cortical interneuron | 90.49 | 89.97 |
| forebrain radial glial cell | 91.68 | 89.86 |
| stratified epithelial cell | 94.34 | 89.60 |
| mueller cell | 90.74 | 89.57 |
| pvalb gabaergic cortical interneuron | 90.04 | 89.41 |
| diffuse bipolar 2 cell | 87.44 | 88.83 |
| cerebellar granule cell | 89.83 | 88.81 |
| invaginating midget bipolar cell | 92.54 | 88.31 |
| mesangial cell | 93.77 | 88.14 |
| skin fibroblast | 89.42 | 87.84 |
| neural progenitor cell | 92.58 | 87.72 |
| mesothelial cell | 87.34 | 87.69 |
| preadipocyte | 88.55 | 87.63 |

*Continued on next page*

Table 9 – *Continued from previous page*

| Cell Type | Accuracy (%) | F$_1$ (%) |
|---|---|---|
| central nervous system macrophage | 92.30 | 87.37 |
| flat midget bipolar cell | 92.21 | 87.06 |
| intestinal epithelial cell | 86.30 | 86.75 |
| syncytiotrophoblast cell | 89.08 | 86.69 |
| epithelial cell of lower respiratory tract | 86.86 | 86.65 |
| sst gabaergic cortical interneuron | 86.23 | 85.94 |
| sncg gabaergic cortical interneuron | 89.06 | 85.83 |
| regular ventricular cardiac myocyte | 92.99 | 85.60 |
| vip gabaergic cortical interneuron | 82.23 | 85.58 |
| neural cell | 83.60 | 85.36 |
| iga plasma cell | 94.16 | 85.09 |
| mast cell | 85.06 | 85.01 |
| glycinergic amacrine cell | 82.93 | 84.99 |
| erythroblast | 88.93 | 84.88 |
| gabaergic amacrine cell | 86.47 | 84.82 |
| endothelial cell of lymphatic vessel | 84.92 | 84.51 |
| cerebral cortex gabaergic interneuron | 89.19 | 84.33 |
| purkinje cell | 81.59 | 84.32 |
| basal cell | 84.13 | 84.09 |
| neuron | 75.49 | 83.85 |
| respiratory basal cell | 88.00 | 83.44 |
| enterocyte | 86.32 | 82.39 |
| luminal hormone-sensing cell of mammary gland | 89.44 | 82.31 |
| retinal bipolar neuron | 76.85 | 82.26 |
| ciliated columnar cell of tracheobronchial tree | 94.96 | 81.81 |
| cardiac muscle cell | 77.48 | 81.64 |
| alveolar macrophage | 81.73 | 81.03 |
| fibroblast of mammary gland | 85.48 | 80.24 |
| mesenchymal stem cell | 90.89 | 80.02 |
| neutrophil | 77.70 | 79.91 |
| thymocyte | 87.15 | 79.39 |
| retinal ganglion cell | 71.92 | 79.01 |
| decidual natural killer cell, human | 85.98 | 78.67 |
| amacrine cell | 74.08 | 78.04 |
| mononuclear phagocyte | 82.07 | 77.82 |
| luminal adaptive secretory precursor cell of mammary gland | 85.00 | 77.67 |
| endothelial tip cell | 84.65 | 77.31 |
| elicited macrophage | 86.02 | 77.26 |
| double-positive, alpha-beta thymocyte | 83.73 | 77.14 |
| mesodermal cell | 90.12 | 76.44 |
| astrocyte | 71.34 | 75.32 |
| tracheal goblet cell | 77.14 | 75.23 |
| perivascular cell | 76.84 | 75.00 |
| mammary gland epithelial cell | 73.30 | 74.81 |
| fibroblast of lung | 59.78 | 73.96 |
| ciliated cell | 59.31 | 73.94 |
| microglial cell | 64.81 | 73.79 |
| nasal mucosa goblet cell | 89.43 | 73.68 |
| plasma cell | 66.51 | 73.21 |
| naive b cell | 76.43 | 71.71 |
| endothelial cell of vascular tree | 74.54 | 71.47 |
| cd14-low, cd16-positive monocyte | 78.99 | 71.21 |
| memory b cell | 73.77 | 71.15 |
| secretory cell | 65.63 | 71.15 |
| luminal epithelial cell of mammary gland | 63.87 | 71.05 |
| glutamatergic neuron | 67.95 | 70.91 |
| inhibitory interneuron | 65.37 | 69.87 |
| neuronal brush cell | 75.75 | 68.77 |
| macroglial cell | 68.19 | 68.23 |
| vascular associated smooth muscle cell | 69.99 | 67.75 |
| blood vessel endothelial cell | 62.86 | 67.58 |

*Continued on next page*

Table 9 – *Continued from previous page*

| Cell Type | Accuracy (%) | F$_1$ (%) |
|---|---|---|
| mesenchymal cell | 67.39 | 67.16 |
| progenitor cell | 57.59 | 65.91 |
| double negative thymocyte | 57.32 | 65.21 |
| epithelial cell | 58.02 | 65.20 |
| cd14-positive monocyte | 68.24 | 64.03 |
| stellate neuron | 63.25 | 63.91 |
| mature alpha-beta t cell | 66.56 | 63.67 |
| stromal cell | 55.02 | 63.30 |
| myeloid cell | 60.48 | 62.78 |
| macrophage | 55.73 | 62.50 |
| vein endothelial cell | 68.16 | 62.02 |
| gabaergic neuron | 59.47 | 61.78 |
| smooth muscle cell | 62.76 | 61.40 |
| pericyte | 55.40 | 60.95 |
| helper t cell | 69.61 | 60.52 |
| glial cell | 51.89 | 60.24 |
| myofibroblast cell | 58.39 | 59.66 |
| endothelial cell of artery | 58.84 | 59.58 |
| neuron associated cell (sensu vertebrata) | 55.91 | 59.27 |
| fibroblast | 57.53 | 59.16 |
| endothelial cell | 48.58 | 56.48 |
| club cell | 46.93 | 55.89 |
| conventional dendritic cell | 53.53 | 55.80 |
| dendritic cell | 55.01 | 55.48 |
| capillary endothelial cell | 51.63 | 54.40 |
| granule cell | 64.26 | 54.20 |
| cd16-positive, cd56-dim natural killer cell, human | 56.25 | 53.98 |
| non-classical monocyte | 55.09 | 53.13 |
| cd16-negative, cd56-bright natural killer cell, human | 60.32 | 51.29 |
| leukocyte | 42.43 | 50.08 |
| classical monocyte | 45.91 | 49.66 |
| b cell | 40.68 | 49.18 |
| mucosal invariant t cell | 59.40 | 48.63 |
| mature nk t cell | 39.99 | 47.78 |
| cd8-positive, alpha-beta cytotoxic t cell | 53.13 | 45.89 |
| innate lymphoid cell | 39.14 | 44.94 |
| cd4-positive, alpha-beta memory t cell | 45.64 | 43.70 |
| cd4-positive helper t cell | 41.15 | 41.71 |
| naive thymus-derived cd8-positive, alpha-beta t cell | 41.24 | 38.61 |
| monocyte | 30.16 | 38.25 |
| natural killer cell | 29.91 | 35.49 |
| cd4-positive, alpha-beta cytotoxic t cell | 40.71 | 35.18 |
| cd4-positive, alpha-beta t cell | 32.57 | 32.07 |
| cd8-positive, alpha-beta memory t cell | 28.24 | 32.03 |
| effector memory cd8-positive, alpha-beta t cell | 31.15 | 31.17 |
| regulatory t cell | 26.52 | 30.82 |
| naive thymus-derived cd4-positive, alpha-beta t cell | 43.29 | 30.00 |
| t cell | 22.55 | 29.68 |
| central memory cd8-positive, alpha-beta t cell | 31.73 | 29.39 |
| gamma-delta t cell | 20.02 | 28.83 |
| central memory cd4-positive, alpha-beta t cell | 25.78 | 26.58 |
| effector memory cd4-positive, alpha-beta t cell | 25.01 | 26.49 |
| cd8-positive, alpha-beta t cell | 13.82 | 20.37 |

# N  CELLxGENE: Detailed Performance Results for Tissue Type Classification

We examine whether representations support mesoscale discrimination across tissues. As with cell types, we report per-class macro-$F_1$ in Table 10 to characterize performance under substantial class-size variation, and present the confusion matrix along with per-class support (#cells) in Figure 9. CELLxPERT achieves high $F_1$ for transcriptionally stereotyped tissues, including *central nervous system* (n=8.16M; $F_1$=91.5%), *eye* (n=1.47M; $F_1$=95.9%), *embryo* (n=119,716; $F_1$=94.3%), *adipose tissue* (n=188,861; $F_1$=89.0%), and *yolk sac* (n=43,096; $F_1$=90.2%), reflecting strong, tissue-specific transcriptional programs. Confusions arise primarily where anatomical or functional overlap induces shared signatures: *intestinal system* (n=581,826; $F_1$=66.9%) vs. *mucosa* (n=49,857; $F_1$=70.5%), *lung* (n=1.04M; $F_1$=70.0%) vs. *respiratory system* (n=255,173; $F_1$=69.1%), and *endocrine gland* (n=264,924; $F_1$=80.8%) vs. *adrenal gland* (n=327,928; $F_1$=90.9%) where steroidogenic programs overlap. Similarly, *blood* (n=4.04M; $F_1$=62.9%) frequently overlaps with *lymph node* (n=121,646; $F_1$=64.3%) and *spleen* (n=212,691; $F_1$=55.6%) due to shared immune signatures.

**Table 10:** Classification performance for 30 tissue types

| Tissue | Accuracy (%) | $F_1$ (%) |
|---|---|---|
| eye | 94.33 | 95.85 |
| embryo | 94.04 | 94.34 |
| central nervous system | 95.37 | 91.46 |
| adrenal gland | 88.97 | 90.87 |
| yolk sac | 90.37 | 90.21 |
| adipose tissue | 91.94 | 89.01 |
| uterus | 89.24 | 88.88 |
| limb | 91.97 | 88.47 |
| placenta | 87.20 | 87.25 |
| reproductive system | 82.34 | 84.40 |
| breast | 75.83 | 83.25 |
| esophagus | 82.25 | 83.05 |
| heart | 81.17 | 82.91 |
| nose | 79.56 | 81.45 |
| endocrine gland | 75.59 | 80.80 |
| kidney | 76.36 | 80.46 |
| ovary | 73.22 | 75.74 |
| liver | 72.87 | 75.72 |
| fallopian tube | 77.24 | 75.29 |
| skin of body | 77.88 | 74.66 |
| pancreas | 75.05 | 73.81 |
| musculature | 82.00 | 73.32 |
| mucosa | 60.90 | 70.53 |
| lung | 72.91 | 69.61 |
| respiratory system | 72.33 | 69.13 |
| intestinal system | 55.17 | 66.93 |
| lymph node | 71.88 | 64.33 |
| blood | 77.26 | 62.89 |
| stomach | 61.97 | 62.63 |
| spleen | 52.91 | 55.58 |

