# OpenReview forum: "CellxPert: An Efficient Reasoning Language Model for Single-Cell and Spatial Multi-Omics"
_ICLR.cc/2026/Conference — ICLR 2026 Conference Withdrawn Submission_

### Official Review · Reviewer_M94x · 2025-10-31

**Soundness:** 3
**Presentation:** 3
**Contribution:** 3
**Rating:** 8
**Confidence:** 4

**Summary:**

This manuscript introduces CELLXPERT, a large multimodal foundation model designed for single-cell and spatial multi-omics data. The model unifies diverse assay types (scRNA-seq, scATAC-seq, CITE-seq, MERFISH, imaging mass cytometry) into a shared latent representation, supports cell-type annotation across 154 identities, and enables downstream fine-tuning through low-rank adapters. A central contribution is an in silico perturbation framework based on blockwise Metropolis–Hastings sampling, which iteratively proposes and accepts expression updates so that predicted perturbation responses remain biologically plausible and close to the data manifold. Architecturally, the model incorporates mixture-of-experts layers, expanded token context, and spatial neighborhood encoding. The method is evaluated on perturbation benchmarks, multimodal integration tasks, spatial imaging data, and disease fine-tuning scenarios, where it demonstrates improved performance compared to existing single-cell foundation models such as Geneformer, scGPT, and CellPLM. Overall, the manuscript aims to provide a unified, biologically grounded approach to reasoning over complex cellular states.

**Strengths:**

A notable strength of this work is its conceptual unification of disparate single-cell and spatial modalities into a single latent framework, which is extremely relevant given the rapidly diversifying assay landscape. The perturbation modeling approach is both theoretically motivated and practically useful: by leveraging iterative Metropolis–Hastings masking rather than single-pass imputation, the method avoids off-manifold drift and preserves gene–gene relationships, an issue that has limited existing perturbation simulators. The architectural choices—including mixture-of-experts and long-context reasoning—appear well justified by the hierarchical structure of cellular data. Empirically, the method shows consistent improvements across perturbation benchmarks, multimodal integration, and cell-type classification, and the manuscript provides ablations demonstrating that several architectural pieces contribute meaningfully to performance. The integration of LoRA adapters is also practical, offering domain customization for real disease datasets without catastrophic forgetting. Finally, the spatial results on MERFISH/IMC add evidence that the model generalizes beyond purely transcriptomic inputs, which strengthens its claims of multimodal reasoning.

**Weaknesses:**

While the perturbation sampling strategy is a highlight of the paper, its computational cost relative to single-pass inference is not fully characterized; iterative MCMC steps may limit scalability on large studies or high-target gene screens. The manuscript would also benefit from more analysis of robustness across training seeds, especially for perturbation outcomes, since stochastic sampling procedures can introduce variance. Although the model claims broad multimodal generality, most biological interpretation is still shown in hematopoietic or immune contexts, leaving open how well it handles highly heterogeneous solid tissues. The claims regarding improved biological realism of perturbation responses rely primarily on embedding-space metrics; direct comparison to differential expression profiles or pathway-level ground truth would strengthen the validation. Additionally, while the energy-based interpretation is discussed, the manuscript stops short of providing formal properties such as convergence guarantees or mixing behavior, which would help justify the correctness of the sampler. Finally, although LoRA fine-tuning is appealing, potential interactions between adapters (e.g., composition across tasks) are not explored, which may be relevant for translational workflows.

**Questions:**

1. How sensitive are MH-based perturbation results to initialization, random seeds, or the number of sampling iterations? Can the authors quantify variance?
2. What is the computational overhead of blockwise MH compared to single-pass masked modeling? Are there practical limits on dataset size or number of perturbation targets?
3. How do the authors detect or guarantee adequate mixing of the sampling chain? Is there a termination criterion beyond a fixed step budget?
4. In settings where the target gene is expressed in very rare subpopulations, can the sampler capture niche perturbation responses, or does averaging wash them out?
5. How stable are attention/importance patterns across different model initializations or LoRA adapter configurations?
6. Have the authors evaluated CELLXPERT on tissues with complex spatial gradients (e.g., tumors, brain) where context is not purely discrete?
7. Can LoRA adapters trained on different diseases be composed, or do they interfere with one another when stacked?
8. For well-studied perturbations with known pathway targets, can the authors demonstrate pathway-level expression shifts rather than only embedding-space movement?

---

> ### Author Response · Authors · 2025-12-03
>
> We thank the reviewer for these technically insightful questions. Below we respond point by point to his questions. We hope these clarifications help. In summary, the current manuscript already provides: (i) sensitivity analysis over the key MH temperature parameter and mixing diagnostics, (ii) explicit runtime and scaling characterization of the MH sampler, (iii) architectural details and constraints of the LoRA adapters, and (iv) spatial and expression-space evaluations on complex biological systems.
>
> ---
>
> **1. Sensitivity of MH-based perturbations to initialization, seeds, and number of iterations; variance**
>
> In our MH sampler we have three main hyperparameters, which are the proposal temperature τ, the chain length T, and the random masking schedule for blocks.
>
> * **Proposal temperature.** We explicitly studied the sensitivity of temperature parameter τ in H.4. We sweeping τ over {0.1, 0.5, 1, 2, 4, 5, 8, 10} on the Replogle K562 ISP benchmark, we observe that both Pearson-∆ with respect to the control reference and with respect to the SYSTEMA perturbed reference increase as τ moves from 0.1 to 2 and then enter a broad plateau. Performance is not hypersensitive within this stable regime, and degradation appears only at very low or very high temperatures. We therefore fix τ = 2 in all main experiments as a conservative choice inside this plateau.
>
> * **Chain behavior and effective burn-in.** In H.4.1 we report the MH acceptance ratio over T = 200 iterations for a representative PMF1 knockdown. Acceptance is initially high and then decays to a lower, stable range, consistent with rapid mixing followed by more selective local moves as the chain approaches high-probability regions of the target distribution.
>
> * **Variance quantification.** In the current manuscript we do not report explicit variance across different random seeds or alternative initializations. Instead, our reported metrics are aggregated over many perturbation targets and cells:
>
>   * ISP alignment is summarized by silhouette scores and the fraction of perturbations where ISP embeddings shift toward the perturbed class, both averaged over hundreds of targets and thousands of cells for each target.
>   * Expression-space performance on SYSTEMA is reported via mean Pearson-∆ across perturbations.
>
>   These averages inherently reduce Monte Carlo variance from the sampler.
>
> ---
>
> **2. Computational overhead of blockwise MH**
>
> In our revised manuscript, we added the computational profile of blockwise MH in H.5. On a single NVIDIA H100 NVL, our blockwise MH sampler runs at 8 iterations per second, requiring 25 seconds per chain with 200 iterations for a batch of 256 control cells. Covering the full high-confidence Replogle subset (with 290 perturbation targets, each requiring 10 to 15 such batches) therefore takes approximately 24 hours of compute. Since ISP sampling is purely inference-time, this overhead is modest relative to pretraining or fine-tuning, and the total cost scales linearly with the number of control cells per line (about 2.5k in Replogle, processed in batches), the context length (up to 4096 tokens), and the number of MH steps $T$.
>
> ---
>
> **3. Mixing diagnostics and termination criterion**
>
> We use standard MCMC diagnostics to assess mixing:
>
> * **Acceptance-ratio trajectories.** Figure 11 plots the per-iteration acceptance ratio for a representative perturbation. The initial high acceptance indicates rapid exploration, followed by a lower but stable acceptance regime as proposals become small local moves in high-probability regions, consistent with convergence of the chain. In the experiments reported in the paper we use a fixed step budget T = 200 for ISP sampling, chosen to be safely beyond the regime where acceptance ratios have stabilized in our diagnostics. We currently do not deploy an adaptive stopping rule based on online convergence checks but instead validate that T is sufficient via these offline diagnostics.

---

> > ### Author Response · Authors · 2025-12-03
> >
> > ---
> >
> > **4. Niche responses in very rare subpopulations**
> >
> > The method is designed specifically to avoid washing out structured, non-global perturbation responses. Replogle-Weissman Perturb-seq screens used in our experiments already have <100 cells per perturbation. During fine-tuning we boost rare classes using minority oversampling, noise injection around bin boundaries, and token-level CutMix between same-label cells in order to mitigate averaging artifacts.
> >
> > * **Anchor medoids instead of centroids.** Additionally, for ISP prediction, we do not anchor the target distribution with a single centroid that could be dominated by common subtypes. Instead, we construct top-K Fréchet **medoids** for each perturbation class, which are actual observed cells that minimize within-class distances and capture multimodal structure by selecting multiple representative modes (My = {m_{y,1}, …, m_{y,K}}).
> >
> > ---
> >
> > **5. Stability of attention/importance patterns across model initializations and LoRA configs**
> >
> > We do not in this version perform a dedicated study of variability of attention maps or importance scores across different random initializations of CELLXPERT, or systematic comparisons of such patterns across alternative LoRA ranks or adapter placements.
> > While we cite generic Transformer explainability work (e.g., Chefer et al.) as compatible tooling, the manuscript does not include attention-based or gradient-based importance visualizations for CELLXPERT itself.  Thus we cannot claim attention/importance stability across seeds or LoRA configurations.
> >
> > ---
> >
> > **6. Evaluation on tissues with complex spatial gradients (tumor, brain)**
> >
> > Yes. CELLXPERT is explicitly evaluated on two spatial datasets with tissues that have complex spatial gradients (tumors, brain) where context is not purely discrete:
> >
> > * **MERFISH mouse brain:** We use 73,655 cells from high-resolution 3D MERFISH, with precise (x, y, z) coordinates and 161-dimensional feature vectors (155 gene targets + controls) spanning 16 annotated cell types, including multiple endothelial, oligodendrocyte, and neuronal populations.
> >
> > * **IMC breast tumors:** We analyze ~1.7×10⁶ cells from imaging mass cytometry of 720 tumor images from 352 patients, representing heterogeneous tumor and stromal microenvironments.
> >
> > Figure 12 shows that CELLXPERT embeddings recover coherent spatial organization for both the brain and tumor tissues, and that Moran’s I varies smoothly across scales, capturing both discrete clusters and continuous gradients in tissue context.
> >
> > Thus, the spatial evaluation already covers settings where context is not purely discrete and includes both complex tumor microenvironments and spatially structured brain tissue.
> >
> > ---
> >
> > **7. Composability of disease-specific LoRA adapters**
> >
> > In the current manuscript, we only instantiate and train a single Parkinson's Disease specific LoRA adapter on the Kamath et al. midbrain dataset using a leave-one-control/one-PD-donor-out protocol. We do not train or stack multiple disease-specific LoRA adapters.
> >
> > ---
> >
> > **8. Pathway-level expression shifts vs. embedding-space movement**
> >
> > We agree that pathway-level analyses are an important complement to embedding-space metrics. The current manuscript already evaluates ISP in **expression space**, but not at the level of curated pathways.
> >
> > In our revision we added the SYSTEMA benchmark (Table 5), where we report mean Pearson-∆ between predicted and ground-truth perturbed expression profiles, under both the standard control-referenced definition and the stricter SYSTEMA perturbed-reference definition. CELLXPERT substantially improves these expression-space metrics over CPA, GEARS, SCGPT, and the perturbed mean baseline. These metrics are computed on gene-level expression and therefore implicitly capture pathway-level alignment.
> >
> > ---

---

### Official Review · Reviewer_7kpK · 2025-11-03

**Soundness:** 3
**Presentation:** 3
**Contribution:** 3
**Rating:** 6
**Confidence:** 4

**Summary:**

The authors of this paper introduce CELLXPERT, a large-scale multimodal foundation model designed to unify single-cell and spatial omics within a shared latent representation. The model integrates scRNA-seq, ATAC-seq, and CITE-seq data, while incorporating spatial omics. CELLXPERT employs an encoder–decoder MoE architecture with a Metropolis–Hastings (MH) sampler for in-silico perturbation (ISP) reasoning, which mitigates out-of-distribution shifts resulting from naïve token deletions or reorderings. This method interprets the masked language model as an implicit energy-based model and samples transcriptomic states iteratively to produce biologically interpretable perturbation trajectories.
Empirical evaluations on multiple benchmarks and spatial datasets show that CELLXPERT outperforms previous single-cell foundation models in perturbation response prediction, cell-type annotation, and multi-omic integration.

**Strengths:**

- The MCMC-based ISP modeling is novel in the field of single-cell foundation models, providing a probabilistic framework that corrects limitations of token-level perturbation approximations in prior models.

- CELLXPERT effectively handles transcriptomic, chromatin, proteomic, and spatial modalities in a unified architecture.

- Across diverse benchmarks, the model achieves superior accuracy and F1 scores, especially on complex multimodal integration tasks.

**Weaknesses:**

- A key limitation is the absence of comparisons to traditional statistical or linear models that have proven highly competitive in recent benchmarking efforts. Prior studies (e.g., Ahlmann-Eltze et al., 2025) demonstrated that simple linear or regression-based models can match or even outperform recent single-cell foundation models on perturbation response prediction tasks under rigorous evaluation metrics.

- Several recent works (e.g., GEARS, MultiVI-GNN) incorporate known gene–gene or spatial relationships using graph neural networks and have shown strong performance in similar perturbation or integration tasks. These methods are only briefly mentioned, leaving unclear how CELLXPERT performs against models that explicitly encode biological priors.

**Questions:**

- Did you evaluate CELLXPERT against such non-deep-learning baselines (e.g., ridge regression, linear mixed models, PCA- or kNN-based predictors)?

- The paper mentions joint modeling of RNA, ATAC, CITE, and spatial data. Did the authors try to use specialized pretrained encoders for these modalities?

- Graph-aware models like GEARS or MultiVI-GNN explicitly encode gene–gene interactions. Could CELLXPERT benefit from incorporating such priors?

---

> ### Author Response · Authors · 2025-12-03
>
> We sincerely thank the reviewer for these thoughtful comments and for highlighting the importance of strong classical baselines and biologically informed models. We address each point below.
>
> ---
>
> ### Comparisons to traditional statistical and linear models
>
> We agree that it is important to benchmark against non–deep learning methods, in line with Ahlmann-Eltze et al. (2025). In the current version we already include such baselines on the **PBMC68K class imbalanced** benchmark, a challenging RNA only cell type classification task with four closely related T, B, and progenitor populations.
>
> On this benchmark we compare CELLXPERT to tree based models, linear models, and scANVI under the same features and splits. The results (Table 2 in the paper) are:
>
> | Model         | Accuracy (%) | Macro-F1 (%) |
> | ------------- | -----------: | -----------: |
> | CELLXPERT     |         78.1 |         78.9 |
> | XGBoost       |         73.4 |         74.2 |
> | Random Forest |         73.0 |         73.4 |
> | L1 logistic   |         68.2 |         69.5 |
> | L2 logistic   |         67.3 |         67.9 |
> | scANVI        |         68.2 |         68.3 |
> | PCA plus kNN  |         68.7 |         67.5 |
>
> These results show that CELLXPERT clearly improves over strong linear and tree based baselines on subtle, imbalanced RNA only cell type distinctions.
>
> For **perturbation response prediction**, our current evaluation focuses on cell level ISP metrics and compares against the main ISP and foundation model baselines, including CPA, GEARS, SCGPT, Geneformer, Geneformer-2, and a perturbed mean baseline (Tables 4, 5, and 6). We did not yet include ridge regression, linear mixed models, or PCA or kNN based predictors for ISP response, since these methods are typically formulated at bulk or summary level rather than in the in silico perturbation setting we study.
>
> ---
>
> ### Relation to graph-aware models and biological priors
>
> We agree that it is important to compare against models that encode biological structure such as gene gene or spatial graphs. In the perturbation benchmarks on **Replogle–Weissman Perturb-seq (K562 and RPE-1)** and **SYSTEMA**, we already include **GEARS**, which is a graph aware model that incorporates gene networks.
>
> On these benchmarks CELLXPERT achieves higher standard Pearson correlations and higher SYSTEMA style Δpert correlations than GEARS on both K562 and RPE-1, and also achieves larger plus Shift fractions. For example, in Table 5 of the paper:
>
> * GEARS reaches standard Pearson Δ of 0.22 (K562) and 0.48 (RPE-1), and SYSTEMA Pearson Δ of 0.00 (K562) and 0.19 (RPE-1).
> * CELLXPERT reaches standard Pearson Δ of 0.66 (K562) and 0.72 (RPE-1), and SYSTEMA Pearson Δ of 0.45 (K562) and 0.46 (RPE-1).
>
> These results indicate that CELLXPERT can match or exceed a strong graph based baseline even without an explicit gene graph in the architecture. CELLXPERT does not explicitly run message passing over a fixed gene or spatial graph. Instead, it uses a transformer with learned attention patterns over quantile binned expression and other modalities. Graph based priors could naturally be incorporated into this framework, for example by using known gene networks to inform attention masks. We consider this a promising extension for future work.
>
> ---
>
> ### Use of specialized pretrained encoders
>
> The paper focuses on a **single unified CELLXPERT backbone** that jointly models RNA, ATAC, CITE, and spatial information in one foundation model. In this work we do not use separate specialized pretrained encoders for individual modalities. Instead, we map all modalities into a shared token space and train CELLXPERT end to end, which allows us to study cross modal transfer and reuse of one model across RNA only, multi omic, perturbation, and spatial tasks.

---

### Official Review · Reviewer_RVXq · 2025-11-06

**Soundness:** 3
**Presentation:** 1
**Contribution:** 2
**Rating:** 2
**Confidence:** 3

**Summary:**

CELLXPERT is a multimodal transformer for single-cell and spatial omics with an MCMC inference scheme for realistic perturbation simulation. It achieves strong results on multiple benchmarks, though its main contribution lies in inference rather than backbone innovation.

**Strengths:**

- The inference approach for **perturbation simulation** is a creative and well-motivated idea that directly targets known limitations of prior token-based perturbation schemes.

- The paper extends **multimodal modeling** to include spatial omics, broadening the scope beyond RNA-only foundation models.

- **Empirical results** are strong, and the ablations on modality fusion are informative.

**Weaknesses:**

- The architectural novelty is limited to engineering refinements, and so similar to previous works. The conceptual novelty lies instead in the proposed perturbation simulation method, which operates at the inference stage and is largely independent of the backbone design.

- As a foundation model, CELLXPERT is expected to demonstrate broad generalization across datasets and modalities. Many standard public datasets used in prior works such as scGPT are not covered in this paper, limiting the assessment of generalization.

- The main novelty (the MCMC inference) remains under-evaluated. Its impact is not isolated from the backbone in Table 3, and comparisons to simpler inference methods are missing. This perturbation simulation method should be applied to embeddings from other foundation models and compared against previously used perturbation approaches under identical setups.

- Several figures lack clarity and proper referencing. ّn Figure 1, the legend text is too small, axis labels are missing, and the subfigures do not show consistent point distributions as implied by the caption, which suggests identical embeddings colored by different labels. Figures 1, 2, 6, and 10 are not referenced or discussed in the main text, leaving their interpretive role unclear.

- No public code or model is provided, which limits verifiability for a foundation model.

Justification: The paper makes interesting steps toward multimodal and perturbation-aware single-cell modeling, but the claims are not sufficiently supported. The foundation backbone adds little architectural (engineering) novelty and does not cover many datasets used in previous works, making it hard to judge generalization. The proposed perturbation simulation approach is novel but not evaluated independently of the backbone. Addressing these points would require new experiments and goes beyond minor revisions, though the direction is promising for future work.

**Questions:**

1. Can you apply the MCMC-ISP wrapper to prior encoders (scGPT, Geneformer) and/or run token-editing/one-shot inference on CELLXPERT to isolate contributions?

2. Report MCMC convergence diagnostics: acceptance-rate curves, stabilization trends, per-chain variance, iteration/compute budgets.

3. Clarify anchor construction, train-split restriction (to avoid leakage), number of medoids (K), and sensitivity to distance choice (Wasserstein vs. cosine/L2).

---

> ### Author Response · Authors · 2025-11-27
> **Fully AI Generated Review & Harsh Scoring**
>
> Dear Area Chair,
>
> I would like to raise a concern about one of the reviews for our submission.
>
> One of the reviews appears to be low-quality, fully AI-generated, and offers a very harsh numerical score without providing the level of detail or technical engagement expected in ICLR reviews. As an example of the issue, similar AI-generated patterns have been documented publicly: https://iclr.pangram.com/reviews?submission_number=23728. While we cannot and do not make assumptions about the reviewer’s identity or intentions, the style and content raise concerns about whether the review meets conference guidelines for constructive, expert feedback.
>
> We respectfully request that you examine this review for adherence to the conference’s expectations around technical depth, fairness, and whether it reflects meaningful human evaluation of the work.
>
> We are actively preparing our rebuttal to address the critics of our reviewers, but we would appreciate the reviewer being reminded to engage more carefully with the submission so the discussion phase can be productive and fair.
>
> Thank you very much for your attention and guidance.

---

> > ### Comment · Area_Chair_Mx16 · 2025-11-27
> >
> > Thank you for raising these concerns, which I take seriously. The website you linked suggests that this review is lightly edited by AI, while the other more positive reviews are considered AI generated. Our internal systems did not raise any alerts for the review you are referring to but for the two other ones. I am in contact with the reviewers to ask for more clarifications and will weigh all reviews fairly and independently.

---

> > > ### Author Response · Authors · 2025-11-27
> > >
> > > Thank you so much for your quick response and for taking our concerns seriously. We really appreciate you contacting the reviewers for more details and promising a fair review process.
> > >
> > > About that linked website, iclr.pangram.com, after looking again, it labels the review by Reviewer RVXq as lightly AI edited instead of fully AI generated. That align with what you said. We're sorry if our first comment made it sound worse than it was from a quick read. Our goal is to point out possible problems with the review's depth and fairness, no matter where it comes from. That said, we still have reservations about its overall quality as constructive feedback, particularly in light of some factual inaccuracies and under engagement with the technical details of our work. For instance:
> > >
> > > - The review says there is limited architectural novelty. This completely misses our key components like hierarchical abstractions across molecular, cellular, and multicellular layers. We also have early multimodal fusion through interleaved tokenization of RNA, ATAC, ADT, and spatial data. Plus additive fusion of identity and magnitude embeddings. And integration of Sparsely-Gated Mixture-of-Experts for more capacity. Dynamic vocabulary growth for unseen genes or modalities. And relative positional encodings for spatial omics. All these go way beyond most prior works, which focus on RNA and skips spatial or multimodal aspects.
> > >
> > > - It criticizes the lack of evaluation on "many standard public datasets used in prior works such as scGPT," yet our benchmarks include key overlapping datasets such as the BMMC multi-omic integration benchmark from OpenProblems NeurIPS 2021 (GEO GSE194122) and Perturb-seq datasets from Replogle-Weissman on K562 and RPE-1 cell lines. We go further by testing on large-scale pretraining data from CELLxGENE Census, same as scGPT from the same source. And extra modalities and datasets like spatial omics with MERFISH and IMC for breast tumors.
> > >
> > > - The idea to isolate the MCMC inference is a good one. But the review ignores our ablations in Section 4 on Experiments. Those compare the MH sampler to baselines like scGPT, which has mode collapse issues, and token deletion or reordering, which cause OOD shifts. We show better cosine similarity shift metrics 90.6 percent versus 15.6 percent (scGPT).
> > >
> > > - The absence of public code is noted, but as per ICLR guidelines and stated in our reproducibility statement, we are going to release the full codebase, configs, checkpoints, and preprocessing scripts on GitHub upon acceptance. We can't reveal our identity.
> > >
> > > - We are preparing a very detailed rebuttal to address these and other points from all reviewers, including conducting new ablation experiments with the MCMC sampling method for perturbation response prediction, using Systema benchmark for baseline comparison.
> > >
> > > Thank you again for your oversight. We value the integrity of the process and look forward to a productive discussion.

---

> > > > ### Comment · Area_Chair_Mx16 · 2025-11-27
> > > >
> > > > I understand your reservations and will take this into account. To have a productive rebuttal, I would suggest that you provide factual counterarguments (as you did in this message to me) to the reviewer's claims. I value the integrity of this process and my overall goal is to ensure that any paper is treated in a fair and unbiased fashion.
> > > >
> > > > Concerning the AI, I want to raise the point that Pangram _also_ indicates that your comments are "fully AI-generated." I would be remiss if I did not point out that this is an inconsistent stance. Again, I will raise this issue with the reviewers but I would ask you, in the interest of fairness, to disclose LLM usage as well. I care about the papers in my stack but I have no interest in watching two LLMs debate the strengths and weaknesses of the work.
> > > >
> > > > Since this is a public comment, may I remind **all participants** of the [ICLR Policies on LLM Usage](https://blog.iclr.cc/2025/08/26/policies-on-large-language-model-usage-at-iclr-2026/) as well the [ICLR Code of Ethics](https://iclr.cc/public/CodeOfEthics).

---

> ### Author Response · Authors · 2025-12-03
>
> We thank the reviewer for emphasizing the importance of broad generalization and for the detailed questions about our MCMC inference, diagnostics, reproducibility, and anchor construction. Below we clarify these points thoroughly.
>
> We already evaluate CELLXPERT across multiple public datasets, modalities, and baselines, as detailed in the main text and appendix. On **PBMC68K class imbalanced** (RNA only cell type classification), we compare CELLXPERT to XGBoost, Random Forest, L1 and L2 logistic regression, PCA plus KNN, and scANVI in Table 2. On the **BMMC CITE-seq** benchmark, we evaluate both a transcriptomics only setting and a full multi omic RNA plus ADT setting, and compare to scMamba, scGPT, and CellPLM in Figure 5b and Table 7. On **Replogle–Weissman Perturb-seq** (K562 and RPE-1), we study genetic perturbation response using standard ISP metrics and the stricter SYSTEMA benchmark, and report clustering and “+ Cosine Shift” metrics against CPA, GEARS, SCGPT, Geneformer, Geneformer-2, and a perturbed mean baseline in Tables 4, 5, and 6. Finally, in the **spatial transcriptomics** datasets (MERFISH and Imaging Mass Cytometry, Appendix I.3) we combine RNA with spatial coordinates to assess classification and downstream spatial analyses such as neighborhood enrichment and spatial autocorrelation. Together, these settings probe generalization across RNA-seq, multi omic, Perturb-seq, and spatial regimes (across different platforms including MERFISH and Imaging Mass Cytometry) using the main foundation model baselines from prior work.
>
> Regarding the novelty of the MCMC inference, the MCMC ISP wrapper is conceptually model agnostic and can be applied to any single cell foundation model trained with a masked language modeling objective, including scGPT and Geneformer. In practice, integrating and validating a new sampling based inference layer into multiple external codebases is nontrivial within this revision. To directly address the reviewer’s request to isolate contributions, we therefore perform a controlled ablation on the **Replogle–Weissman Perturb-seq K562 and RPE-1** benchmarks, where we keep the CELLXPERT backbone and training fixed and vary only the inference strategy. We compare:
>
> * Iterative blockwise Metropolis–Hastings sampling (our proposed MCMC ISP)
> * One shot masked language model imputation
> * Deterministic token editing that reorders discretized gene expression ranks or bins
>
> The results are:
>
> | Inference method                    | K562 +Shift n (↑) | K562 +Shift % (↑) | RPE-1 +Shift n (↑) | RPE-1 +Shift % (↑) |
> | ----------------------------------- | ----------------- | ----------------- | ------------------ | ------------------ |
> | Metropolis–Hastings (MCMC)          | 212               | 94.6              | 64                 | 97.0               |
> | One shot masked imputation          | 171               | 76.3              | 47                 | 71.2               |
> | Token editing (rank or bin reorder) | 140               | 62.5              | 41                 | 62.1               |
>
> **Table 6:** Inference strategy ablation on CELLXPERT for ISP response prediction on Replogle K562 and RPE-1.
>
> Metropolis–Hastings sampling yields substantially larger target directed cosine shifts on both cell lines. On K562, it reaches 94.6 percent positive shifts, a gain of 18.3 points over one shot imputation and 32.1 points over token editing. On RPE-1, it reaches 97.0 percent, improving by 25.8 and 34.9 points. This controlled ablation shows that, with the backbone and training held fixed, Bayesian MCMC ISP inference is the primary driver of the gains in ISP response prediction. Applying the MCMC wrapper to other encoders such as scGPT and Geneformer is a natural next step, but is outside the scope of the current revision.

---

> ### Author Response · Authors · 2025-12-03
>
> Regarding MCMC convergence diagnostics and compute budget, we already report the requested information and will cross reference it more clearly. Appendix H.4.1 and Figure 11 show the Metropolis–Hastings acceptance ratio in the range [0, 1] as a function of iteration for a representative PMF1 knockdown ISP run. The curve starts with high acceptance rates, indicating fast initial mixing, and then stabilizes as proposals become smaller, consistent with convergence of the Markov chain. Appendix H.5 reports the practical MCMC budget and runtime. For each batch of control cells we run a blockwise Metropolis–Hastings chain at inference time using encoder forward passes only. At each iteration we mask a random block of non target genes and evaluate two MLM forward passes, so the cost is linear in batch size, sequence length, and number of MCMC steps. On a single NVIDIA H100 NVL, our blockwise MH sampler runs at 8 iterations per second, requiring 25 seconds per chain with 200 iterations for a batch of 256 control cells. Covering the full high-confidence Replogle subset (with 290 perturbation targets, each requiring 10 to 15 such batches) therefore takes approximately 24 hours of compute. Since ISP sampling is purely inference-time, this overhead is modest relative to pretraining or fine-tuning, and the total cost scales linearly with the number of control cells per line (about 2.5k in Replogle, processed in batches), the context length (up to 4096 tokens), and the number of MH steps $T$.
>
> On reproducibility, as detailed in our Reproducibility Statement, all datasets used in our experiments are publicly available, and we provide direct links and complete preprocessing scripts in our anonymized codebase. Due to the double blind review process we cannot release the full repository, configuration files, and checkpoints at this stage without compromising anonymity. Upon acceptance, we will release code, configuration files, pretrained checkpoints, and a detailed README with setup and evaluation instructions so that all experiments can be reproduced end to end.
>
> Finally, regarding anchor construction and sensitivity, anchor cells are always constructed only from the training split, which prevents any leakage from test data. For each perturbation, we embed all training cells under that perturbation and select a small set of representative anchor medoids in this embedding space. These anchors define the perturbation specific target distribution that CELLXPERT uses at inference time. No test cells and no embeddings computed from test cells are used in anchor selection or in shift metrics. We conduct a sensitivity analysis over the number of medoids K and observe that ISP and shift metrics remain stable across a reasonable range of K, with only minor changes compared with the gains from the inference procedure. This indicates that the method is not sensitive to the exact choice of K as long as each perturbation has enough anchors to capture its main modes. Additionally, we compare Wasserstein-1 to cosine and L2 distances. Empirically, Wasserstein-1 gives consistently better ISP response prediction and cosine shift metrics. Conceptually, this is expected, since Wasserstein-1 is an optimal transport distance that is sensitive to shifts in the full distribution of expression, not only to changes in the mean, whereas cosine and L2 focus on pointwise differences and are less informative about distributional shape. This is why we adopt Wasserstein-1 as our primary distance for anchor construction and scoring.

---

### Official Review · Reviewer_DDYi · 2025-11-10

**Soundness:** 1
**Presentation:** 3
**Contribution:** 2
**Rating:** 2
**Confidence:** 4

**Summary:**

The paper introduces CellxPert, a new single-cell foundation model that uses a Transformer encoder-decoder trained with a masked-language modeling objective and sparsely gated Mixture-of-Experts feed-forward layers. The model allows for incorporating multi-omic measurements, where each feature is assigned an identity embedding and a discretized bin embedding based on its magnitude of activity in a given cell. Additionally, the model accommodates spatial transcriptomics-based positional encoding, when such data is available. A key distinction from existing single-cell foundation models is their approach to perturbation modeling, where, rather than performing one-shot imputations, they adopt a Metropolis-Hastings MCMC approach. This approach repeatedly masks a small block of (non-directly-perturbed) genes and proposes updates to impute them based on the encoder's conditional distributions and accepts proposals with probability proportional to how closely they move the cell profile to perturbed-cell anchors.

While there are several interesting ideas in the proposed model that could potentially advance the state of the single-cell foundation models, the evaluation rigor is lacking, leaving it unclear whether these ideas actually provide any advances (thus my current rating on Soudness), and certain key modeling choices appear underpowered.

**Strengths:**

**1.** The submission proposes a single-cell foundation model that incorporates multi-omic data. This is a biologically well-motivated and timely problem with very little development, thus, has a promising direction.
**2.** The identity-plus-binned-magnitude scheme (with permutation training for order invariance) is biologically sensible. They also have practical engineering choices with, e.g., sparsely-gated MoE layers.
**3.** The authors rightly identify some of the problems with the approaches current single-cell foundation models take to simulate perturbations (particularly the one-shot imputation approaches and OOD issues) and propose a new approach to improve upon these.
**4.** There is a range of datasets used for experimental benchmarking, involving various types of -omics data, clinical datasets with patient-hold-out experiments, and genetic perturbations.
**5.** Introduction and Related Works sections are well-written and motivated.
**6.** The method is generally clearly described although some experimental details may be lacking.

**Weaknesses:**

**1.** The largest concern I have is the lack of appropriate benchmarking that clearly establishes whether the proposed model is a stronger foundation model than what already exists in the single cell literature (at least for transcriptomics). There are only two experiments where other single-cell foundation models (scFM) are used as baselines, and both have “confounding factors” for interpretation:
&nbsp;&nbsp;&nbsp;&nbsp; **(i)** Genetic Perturbation Clustering Task: Since the submission introduces both a new FM architecture *and* a new perturbation simulation approach (Metropolis-Hastings scheme), it is unclear which is responsible for the performance difference in the genetic perturbation tasks since such ablations are missing.
&nbsp;&nbsp;&nbsp;&nbsp; **(ii)** Multi-omic Cell-Type Annotation Task: In the multi-omic case, the scFM baselines are transcriptomics-only models. So, I believe the authors compare the performance of their model when using multi-omic data vs the performance of the scFM models when they only use transcriptomics data. Unless I misinterpret the writing, there are no results on CellxPert classifying BMMC cells with transcriptomics only. This makes it unclear if the performance gain is from the multi-omic data or the model architecture. A typical benchmarking experiment used by existing foundation models is to test cell type annotation performance under the same conditions (e.g. with transcriptomics, like in Section 4 paragraph 2). I don’t believe we, as a field, have figured out the “right benchmarks” yet, but I'd expect to see such comparison.

**2.** Most tasks involve classification but there is no linear classifier baseline. This is important because one of the main criticisms in the field is that the current foundation models fail to robust outperform linear models in most tasks *(e.g. Systema paper by Vinas Torne et al, Nature Biotechnology 2025; Boiarsky et al Nature Machine Intelligence 2024 etc)*. Such a baseline should always be included in ML papers.

**3.** On a similar note, it is not clear in the spatial transcriptomic + proteomic experiments whether the incorporation of spatial coordinates as positional encodings actually provides anything useful to the model, e.g. I would expect to see a comparison on cell type annotation when these coordinates are completely ignored.

**4.** While I excitedly appreciate the effort to move towards incorporating multi-omic data, the current design appears underpowered. The model enforces a fixed per-cell token budget shared across all modalities. A typical single-cell ATAC-seq data involves tens of thousands peaks and often resolves cell states that transcriptomics does not *(e.g. Jindal et al Nature Biotechnology 2023)*, but the current modeling approach will discard most of the features that would explain a large portion of the variability. In its current state, the multi-omic fusion feels naïve. Additionally, there is no principled token-allocation policy across modalities so it is unclear how a user would go about determining how many tokens to allocate to one modality vs the other. Lastly, Figure 5 results are demonstrated as if RNA features are more informative than ATAC features but it is unclear if the model was pre-trained on ATAC (or ADT) data at all. I believe the 2023 version of CELLxGENE only has transcriptomic data, so it would not be surprising if the model did not learn as informative embeddings for ATAC and ADT  features, for example.

**5.** One of the goals of scFM field is to eventually simulate effects of unseen perturbations. While we are far from this goal, it is unclear how one would simulate perturbation profiles when there is no data to pick anchors for the Metropolis-Hastings procedure. Additionally, a few key information is missing in this section:
&nbsp;&nbsp;&nbsp;&nbsp; **(a)** It is unclear how the temperature parameter \tau (for exploration vs exploitation tradeoff in proposals) and the \beta parameter in Wasserstein-1 objective are picked and whether the perturbation simulation performance is sensitive to these.
&nbsp;&nbsp;&nbsp;&nbsp; **(b)** A fair comparison would require that scFM baselines are fine-tuned on the training split since training split is used for anchors in CellxPert. I assume this is the case but I could not locate any information on this in the paper.
&nbsp;&nbsp;&nbsp;&nbsp; **(c)** Some baselines used with K562 cells are not used with RPE-1 cells and it is unclear why, unless I am missing a note in the manuscript.
&nbsp;&nbsp;&nbsp;&nbsp; **(d)** For cosine-shift calculations, which cells' embeddings are used: training split, test split, all? This is quite important to clarify since depending on the choice, you may be coupling the benchmark/metric to the model itself in some ways.

From the point of view of my own evaluation, I think the most reasonable and effective way to improve the submission would be to include appropriate, robust benchmarking experiments that address concerns #1-#3. I appreciate the effort toward multi-omic scFM and believe there are some interesting ideas in the manuscript, so I would be happy to raise my score with demonstrated improvements regarding these concerns.

**Questions:**

**1.** How do the authors propose users to determine how many tokens to allocate per modality?
**2.** Is the quantile binning performed on a gene-by-gene basis or are bin boundaries determined on expression levels across all genes?
**3.** Are there any experiments showing whether four random permutations is sufficient to achieve position invariance?
**4.** Are the scFM baselines on Table 3 fine-tuned on the training split?
**5.** For cosine-shift calculations, which cells' embeddings are used: training split, test split, all?
**6.** Do the scFM baselines in Table 4 all only take in transcriptomic data?
**7.** How do the authors propose to simulate perturbation effects with no training data to pick anchors?

---

> ### Author Response · Authors · 2025-12-03
>
> We thank the reviewer for their thoughtful feedback, which helped us substantially improve the paper. Below we explain how the new experiments, ablations, and clarifications address all of the raised concerns.
>
> ---
>
> ### New benchmarks and overall performance
>
> In response to the comments about evaluation rigor and benchmarking fairness, we added two more challenging benchmarks:
>
> 1. PBMC68K class imbalanced (Boiarsky et al., 2024)
> 2. SYSTEMA (Viñas Torné et al., 2025)
>
> On PBMC68K, CELLXPERT attains 78.9% macro F1 versus 74.2% for XGBoost, and also outperforms L1 and L2 logistic regression, PCA plus KNN, and scANVI (Table 2). This shows that our model improves over classical machine learning models and linear baselines on RNA only data.
>
> On SYSTEMA, which evaluates perturbation specific response rather than only a global perturbed versus control shift, CELLXPERT achieves Pearson Δ = 0.45 (K562) and 0.46 (RPE 1) using the strict perturbed reference, while state of the art models such as CPA, GEARS, and SCGPT fall near zero. Thus CELLXPERT captures both the global shift and the residual, target specific expression changes more accurately than prior methods.
>
> ---
>
> ### Benchmark against linear classifiers (PBMC68K)
>
> We evaluated classical RNA only models on the PBMC68K task with four closely related T, B, and progenitor cell types. Table 2 summarizes the results.
>
> | Model         | Accuracy (%) | Macro F1 (%) |
> | ------------- | ------------ | ------------ |
> | CELLXPERT     | 78.1         | 78.9         |
> | XGBoost       | 73.4         | 74.2         |
> | Random Forest | 73.0         | 73.4         |
> | L1 Logistic   | 68.2         | 69.5         |
> | L2 Logistic   | 67.3         | 67.9         |
> | scANVI        | 68.2         | 68.3         |
> | PCA + KNN     | 68.7         | 67.5         |
>
> **Table 2:** PBMC68K class imbalanced benchmark. CELLXPERT outperforms all classical baselines in macro F1. This confirms that linear and tree based models do not match the foundation model on these subtle, imbalanced classes.
>
> ---
>
> ### SYSTEMA benchmark for perturbation prediction
>
> We added a SYSTEMA based benchmark to directly assess perturbation specific prediction. SYSTEMA highlights that standard ISP metrics mostly reward models for capturing a shared perturbed versus control shift, rather than perturbation specific effects.
>
> SYSTEMA recenters on the global perturbed mean (Δpert), so models must explain only target specific residuals. Under this stricter metric, most methods, including the perturbed mean baseline, drop to near random performance.
>
> CELLXPERT remains strong. It keeps high control centered scores and achieves relatively high SYSTEMA style Δpert correlations, around 0.45, indicating substantially better perturbation specific prediction.
>
> | Model               | Pearson Δ (Standard) K562 (↑) | Pearson Δ (Standard) RPE 1 (↑) | Pearson Δ (SYSTEMA) K562 (↑) | Pearson Δ (SYSTEMA) RPE 1 (↑) |
> | ------------------- | ----------------------------- | ------------------------------ | ---------------------------- | ----------------------------- |
> | CPA                 | 0.06                          | 0.10                           | 0.05                         | 0.08                          |
> | GEARS               | 0.22                          | 0.48                           | 0.00                         | 0.19                          |
> | SCGPT               | 0.27                          | 0.51                           | 0.06                         | 0.13                          |
> | Perturbed mean      | 0.32                          | 0.55                           | 0.06                         | 0.08                          |
> | CELLXPERT           | 0.66                          | 0.72                           | 0.45                         | 0.46                          |
> | Gain over next best | +0.34                         | +0.17                          | +0.39                        | +0.27                         |
>
> **Table 5:** SYSTEMA ISP benchmark in expression space on Replogle K562 and RPE 1. We report mean Pearson Δ with the control reference (standard) and with the SYSTEMA perturbed reference.

---

> > ### Author Response · Authors · 2025-12-03
> >
> > ---
> >
> > ### Multi omic cell type annotation (transcriptomics only vs multi omic)
> >
> > Thank you for emphasizing the importance of an apples to apples, transcriptomics only comparison.
> >
> > We apologize that the transcriptomics only setting was not sufficiently highlighted in the original draft. We do report a transcriptomics only version of CELLXPERT on the BMMC benchmark in Figure 5b. The top row, labeled “proteins retained = 0,” corresponds exactly to the gene expression only setting, that is, CITE seq RNA counts without ADTs. When we retain all 3,896 genes and no proteins (top right cell in Figure 5b), CELLXPERT achieves 79.2% Test 1 accuracy using transcriptomics only.
> >
> > This RNA only performance already exceeds all foundation model baselines in Table 7, which report:
> >
> > * scMamba: 77.5%
> > * scGPT: 72.9%
> > * CellPLM: 66.3%
> >
> > Thus even before incorporating additional modalities, CELLXPERT’s transcriptomics only accuracy of 79.2% is higher than all other models in Table 7 under their standard transcriptomics setting. Adding proteins then further improves CELLXPERT from 79.2% (genes only) to 85.7% (genes plus ADTs), isolating the gain from the multi omic signal.
> >
> > ---
> >
> > ### Spatial coordinates
> >
> > Incorporating spatial coordinates as positional encodings did not significantly change classification accuracy in our experiments. However, spatial information enables valuable downstream analyses, including neighborhood enrichment, variogram based spatial autocorrelation, and cell to cell communication inference, as shown in Appendix I.3. These spatial encodings therefore enrich CELLXPERT’s utility in spatial transcriptomics contexts where tissue architecture is important, even if the classification metric itself changes only modestly.
> >
> > ---
> >
> > ### Token budget, multi omic fusion, and temperature ablation
> >
> > In the revised model, we increased the maximum input sequence length from 4,096 to 16,384 tokens. This enables much larger per cell token capacity across RNA, ATAC, and protein modalities.
> >
> > We also introduced bootstrapped permutation encoding, which lets the model ingest the entire set of features from all modalities during training. This removes the need for a fixed token budget or manual allocation policy, so there is no hard limit on how many features per modality are used. As a result, the model can handle high dimensional datasets such as ATAC seq with tens of thousands of peaks, addressing the concern that important regulatory features might otherwise be dropped.
> >
> > We further ran an ablation over the temperature τ that controls the softmax proposal distribution in Metropolis Hastings sampling. On K562, we computed both the standard Pearson Δ, which measures correlation to ground truth perturbed profiles using control as reference, and the SYSTEMA style Pearson Δ, which uses perturbed profiles as reference. Figure 10 shows that both metrics increase as τ rises from 0.1 to about 2.0, then enter a broad plateau. The stricter SYSTEMA Pearson Δpert follows the same trend. ISP performance is therefore not highly sensitive to the exact choice of τ once it lies in this reasonable range. We set τ = 2 in the main experiments as a point in the stable regime that balances exploration and exploitation and avoids the degradation seen at very low temperatures.
> >
> > ---
> >
> > ### Training and evaluation fairness for scFM baselines
> >
> > All scFM baselines (CELLXPERT, Geneformer 2, Geneformer, scGPT) were
> >
> > 1. Trained only on the training split
> > 2. Evaluated only on the test split
> > 3. Given access to the same target perturbations during training, with no leakage of anchor cells or embeddings
> >
> > This matches the CELLXPERT setting, where perturbation anchors are chosen strictly from training cells. We clarify this in Section 6 in the paragraph titled “Target distribution to score new transcriptomic states.”
> >
> > All cosine shift values, which quantify the movement of cell embeddings between unperturbed and in silico perturbed states, were computed exclusively on the test set. We do not evaluate shift on training or anchor cells, which avoids metric contamination.

---

> > > ### Author Response · Authors · 2025-12-03
> > >
> > > ---
> > >
> > > ### Baseline coverage across K562 and RPE 1
> > >
> > > We include Geneformer 2, Geneformer, and scGPT for both K562 and RPE 1. Table 4 summarizes performance on the Replogle–Weissman Perturb seq high confidence subsets. The metrics are mean silhouette across perturbations and “plus Shift,” the count and fraction of targets whose cosine similarity shifts toward the perturbed condition.
> > >
> > > | Model        | K562 Silh. (↑) | K562 +Shift n (↑) | K562 +Shift % (↑) | RPE 1 Silh. (↑) | RPE 1 +Shift n (↑) | RPE 1 +Shift % (↑) |
> > > | ------------ | -------------- | ----------------- | ----------------- | --------------- | ------------------ | ------------------ |
> > > | CELLXPERT    | 0.58           | 212               | 94.6              | 0.63            | 64                 | 97.0               |
> > > | GENEFORMER 2 | 0.61           | 181               | 80.8              | 0.60            | 58                 | 89.0               |
> > > | GENEFORMER   | 0.34           | 65                | 29.0              | 0.45            | 31                 | 47.0               |
> > > | SCGPT        | 0.52           | 35                | 15.6              | 0.57            | 17                 | 25.8               |
> > >
> > > **Table 4:** Replogle–Weissman Perturb seq high confidence subsets. Metrics are mean silhouette across perturbations and “plus Shift,” the count and fraction of targets whose cosine similarity shifts toward the perturbed condition.
> > >
> > > ---
> > >
> > > ### Quantile binning
> > >
> > > The quantile binning is performed across all genes globally, not on a per gene basis. This design captures population level expression magnitude structure. We clarify this in Appendix B.1 of the updated manuscript.
> > >
> > > ---
> > >
> > > ### Fine tuning of scFM baselines
> > >
> > > **Q: Are the scFM baselines in Table 3 fine tuned on the training split?**
> > >
> > > Yes. Thank you for noting the table renumbering. This is now Table 4 in the revised version. All baseline models (Geneformer 2, Geneformer, scGPT) were fine tuned using an 80 to 20 training to test split, matching the CELLXPERT setup. We verified that all models used only the training split for parameter updates and for anchor based operations.
> > >
> > > ---
> > >
> > > ### Few shot perturbation anchors, not zero shot
> > >
> > > **Q: How do the authors propose to simulate perturbation effects with no training data to pick anchors?**
> > >
> > > CELLXPERT follows a few shot perturbation modeling paradigm. It requires a small number of anchor cells per perturbation in the training set to guide inference. The model does not support zero shot simulation of entirely unseen perturbations, which remains an open challenge for the field. Recent benchmarks such as the Virtual Cell Challenge show that even leading models struggle in the strict zero shot regime. Accordingly, CELLXPERT requires access to at least a few anchor cells from the training split to model conditional distributions.

---

> > > > ### Author Response · Authors · 2025-12-03
> > > >
> > > > ---
> > > >
> > > > **(i) Genetic Perturbation Clustering Task**
> > > >
> > > > Thank you for asking to disentangle the effect of the new FM architecture from the new Metropolis–Hastings sampling ISP prediction at inference. To isolate the contribution of the inference strategy from the underlying CELLXPERT architecture and training, we performed an ablation where three decoding strategies are applied to the same pretrained CELLXPERT model:
> > > >
> > > > 1. Iterative blockwise Metropolis–Hastings sampling
> > > > 2. One shot masked language model imputation
> > > > 3. Deterministic token editing that reorders discretized gene expression ranks or bins
> > > >
> > > > The results are summarized in Table 6.
> > > >
> > > > | Inference method                 | K562 +Shift n (↑) | K562 +Shift % (↑) | RPE-1 +Shift n (↑) | RPE-1 +Shift % (↑) |
> > > > | -------------------------------- | ----------------- | ----------------- | ------------------ | ------------------ |
> > > > | Metropolis–Hastings (MCMC)       | 212               | 94.6              | 64                 | 97.0               |
> > > > | One shot masked imputation       | 171               | 76.3              | 47                 | 71.2               |
> > > > | Token editing (rank or bin swap) | 140               | 62.5              | 41                 | 62.1               |
> > > >
> > > > **Table 6:** Inference strategy ablation on CELLXPERT for ISP response prediction on Replogle K562 (n = 224) and RPE-1 (n = 66).
> > > >
> > > > Metropolis–Hastings sampling yields substantially larger target directed cosine shifts than both alternatives. On K562 it reaches 94.6 percent positive shifts, a gain of 18.3 percentage points over one shot imputation and 32.1 over token editing. On RPE-1 it reaches 97.0 percent, improving by 25.8 and 34.9 points, respectively.
> > > >
> > > > These ablations indicate that the gains in ISP response prediction are driven primarily by the Bayesian inference via MCMC sampling, rather than by changes to the model architecture or training procedure.

---

### Author Response · Authors · 2025-12-03
**Thank you**

We thank all reviewers for their careful reading, constructive feedback, and engagement throughout the discussion. We also thank the Area Chair for their hard work. We especially appreciate the detailed suggestions on benchmarking rigor, multi-omic modeling, and the perturbation-inference design.

In particular, Reviewer DDYi wrote that **"I would be happy to raise my score with demonstrated improvements"**. In the revised version, we have addressed nearly all of the raised questions, added the requested linear and classical baselines, included additional benchmarks (e.g., PBMC68K and SYSTEMA), and performed targeted ablations on our MCMC ISP inference and multi-omic fusion strategy. We hope these additions clarify the empirical value of CELLXPERT, and we are grateful for the feedback that aided sharpen our work.

---

### Note · Authors · 2026-01-26

I have read and agree with the venue's withdrawal policy on behalf of myself and my co-authors.

---

### Meta-Review · Area_Chair_KTq7 · 2026-01-07

**Summary:**

The review panel is highly split and a bit dramatic: two reviewers argue the work is promising but under-validated (one clarified an intended borderline score) and are flagged as AI generated contents, while two reviewers recommend rejection due to (i) insufficient and/or mismatched baselines, (ii) concerns about novelty/positioning as “reasoning,” (iii) reliance on embedding-space or potentially confounded metrics, and (iv) limited robustness/generalization and missing practical details (compute, reproducibility).

**Reviewer Concerns:**

### Addressed (in author responses / discussion):

- Authors claim they added/ran additional comparisons, including “near and classical baselines,” and expanded benchmarking to address baseline gaps.

- The paper provides motivation for stricter perturbation evaluation beyond “too-good-to-be-true” Pearson baselines (SYSTEMA framing) and reports improved results under that stricter reference.

### Still outstanding (key decision drivers):

- Baseline/metric adequacy remains disputed: reviewers still question whether the evaluation convincingly establishes gains over simpler baselines and whether embedding-space metrics reflect biological realism.

- Robustness/generalization + cost characterization: limited analysis across seeds, unclear computational cost of sampling, and uncertainty about generalization beyond core contexts remain central concerns.

- Reproducibility: code is not available during review (authors indicate release upon acceptance), which limits verification given the methodological complexity.

**Reviewer Scores:**

Reviewer 7kpK: Likely post-discussion score: stays ~6 (at most a small bump if convinced by added baselines).

Reviewer M94x: initially entered 8, later clarified it was an error and intended 6, listing key remaining concerns (cost, robustness, generalization, embedding metrics, formal properties, LoRA interactions). Likely post-discussion score: 6 (as clarified).

Reviewer DDYi: may move slightly upward if fully satisfied on baselines, but remains below threshold given current record.

Reviewer RVXq: likely unchanged.

---

### Decision · Program_Chairs · 2026-01-26

Reject